# SHARING STATE BETWEEN PROMPTS AND PROGRAMS

**Ellie Y. Cheng    Logan Weber    Tian Jin    Michael Carbin**
MIT CSAIL
{ellieyhc, loganweb, tianjin, mcarbin}@csail.mit.edu

## ABSTRACT

The rise of large language models (LLMs) has introduced a new type of programming: *natural language programming*. Users write prompts, which are instructions in natural language, to direct LLMs to perform tasks such as natural language processing, code generation, reasoning, etc.

An emerging area of research enables interoperability between prompts and programs. We present a novel programming abstraction, *shared program state*, that removes the manual work required to enable interoperability between prompts and program states. With shared program state, programmers can write prompts that directly access program variables, compute with program objects, and implement control flow in the program. We present a schema for specifying *natural function interfaces* that extend programming systems to support programs with prompts and leverage this schema to specify shared program state as a natural function interface.

We implement shared program state in the NIGHTJAR programming system[1]. NIGHTJAR enables programmers to write Python programs containing prompts that share the Python program state. We show that NIGHTJAR programs achieve comparable or higher task accuracy than manually written implementations (+4-19%), while decreasing the lines of code by 39.6% on average. The tradeoff is that NIGHTJAR may incur runtime overhead (0.4-4.3x manual implementations).

## 1 INTRODUCTION

Writing prompts, which are instructions in natural language, for large language model (LLM) systems to evaluate is a new form of programming: *natural language programming*. LLMs have become increasingly capable, leading to widespread adoption of prompting for tasks such as natural language processing, code generation, and mathematical reasoning (Achiam et al., 2023; Chang et al., 2024; Chen et al., 2021; Guo et al., 2025; Liang et al., 2023; OpenAI, 2025; Qin et al., 2023; Wei et al., 2022). Prompts enable a higher level of abstraction in comparison to traditional programs, as prompts require fewer operational details.

**Prompt & Program Interoperability.** Programs offering precision and structure. Programs also support rich programming abstractions, including variables, data structures, and control flow. An emerging trend of programming systems enables *interoperability* between prompts and programs, in which the prompts and the programs operate on common data (Adept, 2024; Anthropic, b; Beurer-Kellner et al., 2023; BoundaryML, 2025; Chase, 2022; Dinu, 2022; Dong et al., 2025; Firebase Team; Gat; Guidance AI; Khattab et al., 2024; Liu & Contributors, 2024; Microsoft, b; Okuda & Amarasinghe, 2024; OpenAI; PrefectHQ; Vaziri et al., 2024; Willard & Louf, 2023; Yuksekgonul et al., 2025; Zheng et al., 2024). Programmers develop programs containing prompts, and these programming systems orchestrate the interface between the prompts and the programs. Figure 1a shows an example Python program containing a prompt.

Existing systems evaluate prompts in a separate environment from the program. Programmers must therefore manually implement the transfer of data and, broadly, *program state* across the boundary between the environments. We term such a design as *isolated program state*. Figure 1a demonstrates the manual work required to transfer program state in the isolated program state design.

---

[1] https://github.com/psg-mit/nightjarpy

```
1   from pydantic import BaseModel
2   from llm_wrapper import llm
3   class Graph: ...
4   class EdgeSchema(BaseModel):
5     src: int
6     tgts: list[int]
7   class GraphSchema(BaseModel):
8     nodes: list[int]
9     edges: list[EdgeSchema]
10  def serialize(g: Graph) -> str: ...
11  def reify(gs: GraphSchema) -> Graph:
12    nodes = set(gs.nodes)
13    edges = {e.src: set(e.tgts) \
14      for e in gs.edges}
15    return Graph(nodes=nodes, edges=edges)
16  def main():
17    graph = Graph(...)
18    while True:
19      query = input("Q: ")
20      class OutputSchema(BaseModel):
21        response: Optional[str]
22        break_flag: bool
23        graph: Optional[GraphSchema]
24      q_out = llm(
25        f"""Perform the <query> with
26  respect to <graph>, where nodes are
27  paper IDs and edges point from a cited
28  paper to a set of papers that cite it.
29  Return `break_flag` as True if <query>
30  indicates termination. Else, return a
31  `response`. If <graph> was updated,
32  return it as `graph`.
33  <query>{query}</query>
34  <graph>{serialize(graph)}</graph>""",
35        output_type=OutputSchema)
36      if q_out.break_flag:
37        break
38      response = q_out.response
39      if q_out.graph:
40        graph = reify(q_out.graph)
41      print(f"A: {response}")
42      print(f"Papers that cite paper 14:")
43      print(graph.edges[14])
44  main()
```

(a) Manual Implementation (Isolated Program State)

```
1   import nightjar
2   class Graph:
3     nodes: set[int]
4     edges: dict[int, set[int]]
5   @nightjar.fn
6   def main():
7     graph = Graph(...)
8     while True:
9       query = input("Q: ")
10      """prompt
11      Perform the <query> with respect to
12      <graph>, where nodes are paper IDs
13      and edges point from a cited paper to
14      a set of papers that cite it. Break
15      if <query> indicates termination.
16      Else, save a <:response> and update
17      <graph> to answer <query>.
18      """
19      print(f"A: {response}")
20      print(f"Papers that cite paper 14:")
21      print(graph.edges[14])
22  main()
```

(b) Example Program with Shared Program State

```
$ ./example.py
Q: Update the graph so paper 5 cites 14.
A: Graph updated.
Papers that cite paper 14:
{ 5 }
Q: Exit, please.
$
```

(c) Example Program Execution Console

Figure 1: Figure 1a shows a program with isolated program state that executes natural language queries on a graph modeling paper citations. Blue highlighted lines show the prompt; gray highlighted lines show the work to transfer the program state between the prompt evaluation and the program. Figure 1b shows the task implemented using shared program state with NIGHTJAR. Figure 1c shows the input and output of an example execution. See Appendix A.1 for the full program.

**Shared Program State.** We present a novel programming abstraction—*shared program state*—in which prompts have direct access to the program state—the variable scopes, the heap, and the control state of the formal program. The prompts read and write program variables (shared scopes), compute with program objects (shared heap), and implement control flow in the program (shared control state). Figure 1b shows an example program written with shared program state.

**Natural Function Interface.** Shared program state and isolated program state are both instances of a *natural function interface*. A natural function interface specifies how prompts interact with the host program execution. We formalized and designed a schema for designing natural function interfaces based on effect and handler program constructs (Bauer & Pretnar, 2015; Leijen, 2016; Plotkin & Power, 2003; Plotkin & Pretnar, 2013). Effects define the actions to be performed, and handlers define the implementations of the actions. Effects and handlers are a generalization of tool use that enables using to implement context switches in system execution.

**Shared Program State Natural Function Interface.** Using our schema, we specified shared program state as a natural function interface. The interface operates over variables, references, labels, and program values. It enables prompts to emit effects to the host language to read and write variables from shared scopes, reference and dereference values from the shared heap, and utilize program labels in the shared control state.

**Contributions.** We present the following contributions in this work:

- We present the *shared program state* abstraction. Shared program state enables developers to write more concise programs and focus on the core program logic. Figure 1a illustrates the extra work a developer would otherwise need to implement themselves to deliver the program with the same functionality as that of Figure 1b.

- We present a schema for specifying *natural function interfaces* that extend programming systems to support programs with prompts. Using this schema, we define the *shared program state natural function interface*, a language-agnostic formal specification that defines the mechanisms by which prompts interact with a host programming system to realize the shared program state abstraction. System developers can use our specification to implement the shared program state abstraction in their own programming systems.

- We implemented the shared program state natural function interface as the NIGHTJAR programming system to demonstrate the efficacy of programming with the shared program state abstraction. We show that NIGHTJAR programs achieve comparable or higher task accuracy than manually written implementations (+4-19%), while decreasing the lines of code by 39.6% on average. The tradeoff to using NIGHTJAR is that it may incur runtime overhead (0.4-4.3x runtime of manual implementations).

**Implications.** Our work introduces shared program state, a new abstraction for programming with prompts in programs. Shared program state enables programmers to focus on program logic by delegating the low-level work of enabling interoperability to the natural function interface implementation. Our work highlights the emerging computing paradigm brought by the rise of LLMs in which computations are dynamically and adaptively planned, implemented, and executed on-the-fly according to the live context.

## 2 RELATED WORK

**Natural Language as Programming.** The idea of integrating natural language in programs dates back to literate programming (Knuth, 1984) and has been popularized by the use of Jupyter and IPython notebooks (Pérez & Granger, 2007). This line of work does not treat natural language as executable code, but instead as documentation describing the behavior of the program or as macros for programmer-written code. The popularity of using natural language prompts to perform computation has risen as LLM capabilities improve. Several works coin prompting as a form of programming (Beurer-Kellner et al., 2023; Feng et al., 2024; Khattab et al., 2024; Liang et al., 2025). Our work takes this a step further by unifying the state between prompts and program.

**Language Interoperability.** *Language interoperability* is the task of enabling a software system to leverage multiple different programming languages computing with common data in its implementation. Language interoperability has a long history (Benton & Kennedy, 1999; Breg et al., 1998; Burroughs Corporation, 1961; Davis et al., 1994; Finne et al., 1999; Graham, 1966; Jeffery et al., 1999; Konstantas, 1993; Laddad & Sen, 2020; Matthews & Findler, 2007; Microsoft, a; Nelson, 1981; Object Management Group, 1991; Oracle, 2017; Osborne et al., 2024; Python Software Foundation, 2025; The MathWorks, Inc., 2025; Wagner et al., 2024), dating back at least 50 years as the software engineering community has sought to build increasingly sophisticated software systems that span increasingly more components, each suited to development in a particular programming languages.

**Isolated Program State.** In recent years, enabling interoperability between prompts and programs have become an area of active research. Most prior work (Anthropic, b; Beurer-Kellner et al., 2023; BoundaryML, 2025; Chase, 2022; Dinu, 2022; Dong et al., 2025; Firebase Team; Gat; Guidance AI; Khattab et al., 2024; Liu & Contributors, 2024; Microsoft, b; OpenAI; Vaziri et al., 2024; Willard & Louf, 2023; Yuksekgonul et al., 2025; Zheng et al., 2024) isolates prompts evaluation from program execution. Specifically, prompts have no visibility to the program state. In general, the programmer writes code to serialize program data to pass to the LLM in the prompt. The LLM returns a string value by default, and the programmer optionally writes a schema to define how the output string is reified into program data. Systems that enable code interpreter tools to answer a natural language prompt (Anthropic, b; Chen et al., 2023; Gao et al., 2023; OpenAI) isolate the code execution from the program that the natural language prompt is embedded in. Our work enables a new programming model of interoperability in which the prompt evaluation shares the program state of the host program.

**Programmer-Defined Tools.** In systems that support interoperable programs with isolated program states, programmers can define custom functions and implement tool use to manually convert LLM responses into operations on the program state. Several systems automatically make tool calls using programmer defined functions (Beurer-Kellner et al., 2023; Chase, 2022; Dong et al., 2025; Firebase Team). Our shared program state abstraction does not require programmers to define custom functions for the prompt to perform operations on the program state.

**Partially Shared Program State.** Other systems enable partially shared program state (Adept, 2024; Dantanarayana et al., 2025; Huang et al., 2023; Okuda & Amarasinghe, 2024; PrefectHQ). Marvin (PrefectHQ) and MTP (Dantanarayana et al., 2025) support reading function arguments and serialization/reification of program data, but do not enable prompts to manipulate mutable objects in-place. AskIt (Okuda & Amarasinghe, 2024) and ANPL (Huang et al., 2023) replace prompts with LLM-generated functions. The generated functions can access object references passed as function arguments. None of the above methods enable prompts to read variables outside of the function scope (and thus cannot support closures), write variables, nor implement control flow. AWL (Adept, 2024) is a domain-specific language for web interactions; prompts evaluate on shared webpages with the program but not shared program state.

## 3   SHARED PROGRAM STATE BY EXAMPLE

NIGHTJAR integrates prompts as first-class code in Python programs: The prompts can read and write Python variables, compute with Python objects, and manipulate the control flow of the surrounding Python program. Figure 1b shows an example program that uses shared program state via NIGHTJAR.

### 3.1   EXAMPLE PROGRAM

The program implements an agent that manipulates a directed graph of paper citations. The nodes are paper identifiers, represented by a set of integers. The edges connect a cited paper to papers that cite it, represented by a dictionary of source nodes to sets of target nodes.

The `main` function uses a `while`-loop (Line 8) to continuously read in user queries from standard input. A triple-quoted string with the language identifier `prompt` demarcates a prompt, as shown on Lines 10 and 18. The prompt specifies the agent's behavior in natural language: The agent performs the given query by writing to a variable named `response` and updating the graph. Execution breaks out of the `while`-loop if the query signals the user is finished with the program; otherwise, the agent prints the response and the papers that cite paper 14 on Lines 19 to 21. Figure 1c shows program outputs with the inputs of a query that adds a new edge and a query that ends the program.

### 3.2   SHARED PROGRAM STATE

At first glance, the prompt in Figure 1b might look like a comment to be filled in by the programmer, but prompts with shared program state are first-class code. Prompts evaluate with shared scopes, shared heap, and shared control state with the Python program state.

**Shared Scopes.** Prompts read and write Python variables in the shared scopes. Prompts can refer to any live Python variables within scope using the syntax ``. For example, on Line 11, the prompt reads the `query` variable, which on the first loop is assigned to `"Update the graph so paper 5 cites 14"`. `<:var>` denotes variable assignments. Line 16 writes a string response `"Graph updated."` to the `response` variable that is printed to standard output on Line 19. Shared scopes enable the `response` variable to be used in Python like any other variable in that scope.

**Shared Heap.** Prompts access Python objects by reference in the Python heap. The prompt in Figure 1b adds an edge in the `graph` object with an in-place update to the existing `graph.edges` dictionary of the existing `graph` object in the program's heap, equivalent to the Python code `graph.edges[14].add(5)`. The update persists after the prompt evaluation. By referencing the existing `graph` pointer, Lines 20 to 21 prints out that paper 5 is in the set of papers that cite paper 14.

**Shared Control State.** Prompts also manipulate the evaluation context in the shared control state through program labels. When the `query` variable indicates the user is finished with the program, like the second example query `"Exit, please"`, the prompt updates the evaluation context to after the

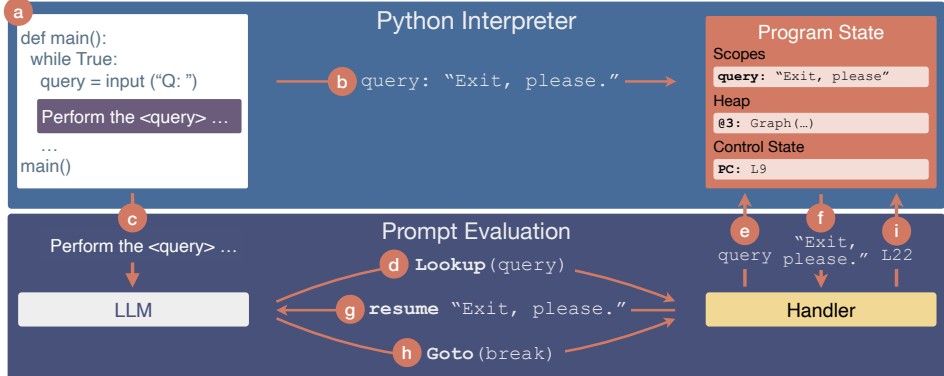

Figure 2: a) An example execution of Figure 1b, in which the prompt uses shared program state. b) The Python interpreter reads and writes to the program state, such as by writing the variable `query`. c) The Python interpreter hands the execution control to the LLM to evaluate the prompt. d) The LLM emits a Lookup request to read the `query` variable. e) The handler reads `query` from the shared scopes, f) which has the value `"Exit, please."` g) The handler resumes prompt evaluation with the value of `query`. h) The LLM emits another request, Goto, to update the evaluation context to the `break` program label (i.e. outside the `while`-loop). Finally, i) the handler manipulates the control state accordingly without resuming prompt evaluation, implementing control flow.

loop, in effect breaking out of it. Correspondingly, the `print`-statements on Lines 19 to 21 are not executed, and the program gracefully exits.

**Without Shared Program State.** Without shared program state, prompts only read and write serialized string values with no other effects on the Python program. Programmers have to write code to manually serialize input data and convert serialized output values into operations on the program state. In Figure 1a, we show an illustrative example of how a programmer might implement the task manually without using shared program state. Without shared program state, the programmer has to write Python code to manually serialize and reify data structures, to update objects in the heap, and invoke control flow primitives. Shared program state provides an abstraction over the intermediary connecting code between a prompt and the program state.

## 3.3 EXECUTING PROGRAMS WITH SHARED PROGRAM STATE

We implemented the NIGHTJAR programming system with shared program state. NIGHTJAR uses an LLM agent to evaluate prompts. The LLM agent issues *effects*, which are requests to the programming system for manipulations to the Python program state. NIGHTJAR implements *handlers*, which receive these effects and interface with the Python program state. Figure 2 shows a diagram of NIGHTJAR evaluating a prompt with shared program state.

Table 1 shows the performance of the Figure 1a program (manual programmer implementation) and the Figure 1b program (using NIGHTJAR) on a graph of 25 nodes with 6 different natural language queries (Appendix A.2). NIGHTJAR achieves higher pass rate in this task than the manual implementation, because the LLM agent adapts to program inputs and inter-

Table 1: Average pass rate and runtime of Figure 1 programs, with time ranges in gray.

| Method | Pass Rate | Time (s) |
|---|---|---|
| Manual Impl | 0.33±0.00 | **31.3** 
 (28.7-35.0) |
| NIGHTJAR (Ours) | **0.83±0.12** | 77.2 
 (72.7-83.0) |

mediate program values. For example, for the query `Update the graph so paper 5 cites 14`, the LLM decides to issue effects to update only the dictionary entry for paper 14. Without shared program state, the programmer must decide and implement the specific mechanisms for the prompt to interact with the program state. In Figure 1a, the programmer implemented updates to the graph by querying the LLM to generate the entire updated graph, which is prone to hallucinations. Shared program state offloads the work of reasoning about and implementing updates to the program state to the LLM agent. The tradeoff to the abstraction is that the LLM agent incurs runtime overhead. Our work shows that engineering efforts reduce the overhead, offering opportunities for future work to improve the accuracy and runtime of shared program state programming systems.

## 4    A NATURAL FUNCTION INTERFACE FOR SHARED PROGRAM STATE

We developed shared program state to enable a higher level of abstraction when using prompts within programs: The prompts manipulate the host program state directly. To define the mechanisms by which a prompt interfaces with the host program state, we first describe the natural function interface schema for extending programming systems with support for prompts in programs. Then, we formally specify shared program state as a natural function interface.

### 4.1    NATURAL FUNCTION INTERFACES

A natural function interface is a specification of how prompts transfer data or utilizes services of a host programming language. A natural function interface provides definitions of 1) the values that cross the boundary between the prompt and the program, 2) *effects*, which are distinguished requests by the prompt for interacting with the surrounding code, and 3) *handlers* that define the behavior of the requests as implemented in the host system. Natural function interface is a generalization of tool use, as defined by OpenAI documentation (OpenAI) and MCP (Model Context Protocol). In Appendix B, we formalized natural function interfaces using effects and handlers (Bauer & Pretnar, 2015; Leijen, 2016; Plotkin & Power, 2003; Plotkin & Pretnar, 2013).

**Values.** The values of a natural function interface are the data types that the prompt takes as input and output from the host system (NFI Values in Figure 3). While LLMs typically take strings as input and output, a natural function interface can specify richer data types and leverage serialization and reification such as through JSON formatting and Pydantic (Colvin et al., 2025) to automatically convert between string and program data types.

**Effects.** The effects of a natural function interface are the requests that the prompt expects the host language to fulfill (NFI Effects in Figure 3). The prompt evaluates to either a value or an effect, denoting the request the host language should fulfill so that the prompt evaluation can continue.

**Handler.** The handler of a natural function interface defines the implementation of the effect requests by the host language (NFI Handler in Figure 3). When the prompt evaluates to an effect, the handler that wraps the prompt catches the effect and fulfills the request.

As described, effects and handlers are similar in behavior to elements of the OpenAI function calling API. In the function calling API, the LLM can request a function to be called and the client of the API will interpret the function and provide the value back to the API, whereupon the LLM resumes its evaluation. Where a natural function interface extends the function calling API is with the ability for the handler to explicitly either resume evaluation (as is typical) or cancel the continued evaluation of the prompt (new). Cancelling the prompt evaluation enables the host program and the prompt to coordinate rich control flow patterns.

A natural function interface is a contract between the host language and the prompt. Specifically, the LLM system that evaluates the prompt must support 1) serialization/reification of the defined values, 2) checkpointing of the prompt evaluation when emitting an effect (as execution control switches to the handlers), 3) restoration of prompt evaluation from a checkpoint when the handler initiates resumption, and 4) cleanup of the prompt evaluation when the handler cancels the prompt evaluation.

### 4.2    SHARED PROGRAM STATE AS A NATURAL FUNCTION INTERFACE

Now we turn to specifying shared program state as a natural function interface, as shown in Figure 3. We assume a general host language that supports variable bindings (shared scopes), mutable state (shared heap), and control flow constructs (shared control state).

**Shared Scopes.** The values to support shared scopes are the names of variables in the host program. The effects are Lookup and Assign. For Lookup requests, the handler retrieves the value of the requested variable in the current scope and uses the $resume$ keyword to restore prompt evaluation with the value. The requested variable must be in the set of variable names the LLM is allowed to read, denoted as $X_i$. For the Assign request, the handler makes the variable binding in the current scope then resumes prompt evaluation with the variable binding still in place. The variable must be in the set of variable names that the LLM is allowed to write to, denoted as $X_o$.

| Variables | $x \in X$ | References | $a \in A$ | Labels | $\ell \in L$ | Host Values | $v$ |
|---|---|---|---|---|---|---|---|

Host Language $\quad e_H ::= v \mid \ldots \qquad\qquad\qquad\qquad$ values
$\qquad\qquad\qquad\qquad \mid x \mid \mathsf{let}\ x = e\ \mathsf{in}\ e \qquad$ variable bindings
$\qquad\qquad\qquad\qquad \mid\ !e \mid \mathsf{ref}\ e \mid e := e \qquad$ mutable state
$\qquad\qquad\qquad\qquad \mid \mathsf{label}\ \ell : e \mid \mathsf{goto}\ \ell \qquad$ control flow

---

| | Prompts | $n$ | Serialization | $\alpha : v \to v_N$ | Reification | $\gamma : v_N \to v$ |
|---|---|---|---|---|---|---|

$\qquad\qquad\qquad$ Prompt Evaluation $\quad \mathsf{nat}_{X_i, X_o, L} : n \times v_N \to v \mid op_N^{X_i, X_o, L}$

NFI Values $\qquad\qquad v_N \subseteq v$

NFI Effects $\qquad op_N^{X_i, X_o, L} ::= \mathsf{Lookup}(x_i) \mid \mathsf{Assign}(x_o, v_N) \qquad x_i \in X_i; x_o \in X_o$
$\qquad\qquad\qquad\qquad\quad \mid \mathsf{Deref}(v_N) \mid \mathsf{Ref}(v_N) \mid \mathsf{Set}(a, v_N)$
$\qquad\qquad\qquad\qquad\quad \mid \mathsf{Goto}(\ell) \qquad\qquad\qquad\qquad\qquad\ \ell \in L$

NFI Handler $\qquad h_N ::= \mathsf{handle}\ \{$
$\qquad\qquad\qquad\qquad\quad \mathsf{finally}\ v_N \to \gamma(v_N);$
$\qquad\qquad\qquad\qquad\quad \mathsf{Lookup}(x) \to resume\ \alpha(x); \qquad$ get variable
$\qquad\qquad\qquad\qquad\quad \mathsf{Assign}(x, v_N) \to$
$\qquad\qquad\qquad\qquad\qquad \mathsf{let}\ x = \gamma(v_N)\ \mathsf{in}\ resume\ \alpha(()); \quad$ set variable
$\qquad\qquad\qquad\qquad\quad \mathsf{Deref}(v_N) \to resume\ \alpha(!\gamma(v_N)); \qquad$ get value at reference
$\qquad\qquad\qquad\qquad\quad \mathsf{Ref}(v_N) \to resume\ \alpha(\mathsf{ref}\ \gamma(v_N)); \quad$ create reference with value
$\qquad\qquad\qquad\qquad\quad \mathsf{Set}(a, v_N) \to$
$\qquad\qquad\qquad\qquad\qquad resume\ \alpha(\gamma(a) := \gamma(v_N)); \qquad$ set value at reference
$\qquad\qquad\qquad\qquad\quad \mathsf{Goto}(\ell) \to \mathsf{goto}\ \gamma(\ell) \qquad\qquad$ set evaluation context to label
$\qquad\qquad\qquad\qquad \}$

Program $\qquad\qquad e ::= e_H \mid h_N(\mathsf{nat}_{X_i, X_o, L}(n, \alpha(v))) \qquad$ prompt with shared program state

Figure 3: Shared Program State Natural Function Interface.

**Shared Heap.** The values to support shared heap are references. Deref, Ref, and Set are the effects. The handler fulfills these effects by resuming prompt evaluation with the value evaluated from the dereference, reference, and set host program construct, respectively.

**Shared Control State.** The values to support shared control state are program labels. The corresponding effect is Goto. The handler dispatches the Goto effect by jumping the evaluation context to the specified label in the effect. Notably, the handler does not resume prompt evaluation while handling this effect, which implements control flow. The label must be in the set of labels that the LLM is allowed to jump to, denoted as $L$.

## 5 NIGHTJAR PROGRAMMING SYSTEM

We implemented the shared program state natural function interface as the NIGHTJAR programming system with Python as the host language. NIGHTJAR uses an LLM agent to translate prompts into effects. In this section, we discuss implementation details for NIGHTJAR.

### 5.1 IMPLEMENTATION

**Shared Scopes.** NIGHTJAR distinguishes Python variables from regular words within a prompt using the syntax `` for variable references and `<:var>` for variable assignments and parses them into $X_i$ and $X_o$, respectively. Specifically, the handler enforces that the LLM agent only lookup ``-denoted variables. If the LLM attempts to look up any other variable, the handler provides the LLM with an error message. This prevents the LLM agent from reading hallucinated variables or variables the programmer does not want the LLM to read. The handler also enforces that `<:var>`-denoted variables must be defined at the end of the prompt evaluation, providing the LLM with an error message if it attempts to terminate evaluation when these variables are undefined. This check enables programmers to reason about which variables are live after a prompt.

**Shared Heap.** NIGHTJAR represents all Python objects as data in the shared heap, represented by references. This enables prompts to manipulate mutable Python objects in-place. Only plain old data types that are immutable, such as strings, integers, floats, booleans, and null-types, are passed directly through the NFI. Additionally, the handler provides the LLM agent with an error message if it attempts to read invalid references.

**Shared Control State.** Because Python does not support the low-level `goto` program construct, NIGHTJAR performs a source code transformation to install `try-except` blocks as program labels. NIGHTJAR exposes program labels corresponding to Python built-in control flow primitives such as `break`, `continue`, `return`, and `raise` to the LLM to use with the Goto effect. NIGHTJAR only exposes program labels that are valid within the evaluation context of the prompt. For example, the program label corresponding to `break` is only exposed when the prompt is within a loop.

## 5.2 SYSTEM ENGINEERING

One implementatin of shared program state in a programming system is by querying the LLM agent to generate the effects exactly as defined in Figure 3. While this implementation is language-agnostic, it is not designed to be performant. Realizing shared program state as a performant abstraction requires additional system engineering. To improve performance, we specialize NIGHTJAR to Python and implement optimizations for reducing runtime.

**Specialization.** We designed shared program state natural function interface to be general to capture the underlying semantics of the shared program state abstraction. The tradeoff of the general approach is the loss of specialized host-language features such as first-class functions, abstract methods, and high-level control flow constructs. It also relies entirely on the LLM to perform data transformations. We specialize NIGHTJAR to Python so that it supports these Python-specific language features and enables Python-based computation. In this implementation, the effects Eval (evaluates a Python expression and returns the result) and Exec (executes a block of Python code in the evaluation context) replace the Lookup, Assign, Deref, Ref, and Set effects. The effects Break, Continue, Return, and Raise replace Goto, corresponding to the respective Python primitives.

**Eager Variable Loading.** During execution, the LLM selects a subset of the variables mentioned in the prompt and inspects their value and type by issuing effects. However, each LLM agent step incurs runtime overhead. We apply an optimization to eagerly load the values and types of the mentioned variables to the LLM, to reduce the number of effects the LLM issues and thus reduce the runtime. The tradeoff is that the value and types of variables the LLM would otherwise not select for inspection are also placed in context, introducing additional token cost.

**Caching.** We implement a simple caching scheme where the LLM API client stores pairs of LLM agent effect trace history and the response. When given identical agent effect trace histories, the client returns the stored response rather than querying fresh responses from LLM APIs. This enables time savings until the LLM reads program inputs it has not seen before.

## 6 EVALUATION

In this work, we presented shared program state, a new programming abstraction that delegates the work of enabling interoperability between a prompt and the program state to the natural function interface implementation. We implemented the NIGHTJAR programming system to execute programs with prompts using shared program state. We next perform an evaluation to answer the following:

- *RQ1.* Does shared program state enable more concise programs?
- *RQ2.* Can programs with shared program state perform as well as manual implementations?

**Benchmarks.** Since there are no existing benchmark suites that evaluate shared program state, we introduce *SPSBench*. *SPSBench* is a suite of 25 programs that contain prompts using shared program state. We adapted several of the programs from the documentation of LLM-related Python packages. We constructed additional programs to evaluate more complex manipulations on the program state by prompts, such as creating closures, in-place state mutations, using generators, defining subclasses, and raising errors. See Appendix C for program descriptions.

**Baselines.** For each program, we implemented the same task using a simplified LLM API that takes a prompt and an optional Pydantic (Colvin et al., 2025) model or JSON schema; we call these *Manual Implementation* programs. *Manual Implementation (Code Interpreter)* executes the manual implementation programs with official code interpreter tools from OpenAI and Anthropic enabled in LLM API calls, which are isolated from the host program but enable the LLM to perform Python-based calculations. The third baseline, NIGHTJAR *(Baseline)*, executes shared program state programs using NIGHTJAR without specialization or optimizations.

Table 2: Average pass rate with standard deviation and average runtime with ranges in gray.

| Model | Method | Pass Rate | Time (s) |
|---|---|---|---|
| Sonnet 4 | Manual Impl | 0.78±0.03 | **8.0** (1.3-44.8) |
| Sonnet 4 | Manual Impl (Code Interpreter) | 0.66±0.04 | 36.8 (3.2-171.5) |
| Sonnet 4 | NIGHTJAR (Baseline) | 0.60±0.04 | 55.6 (9.6-721.0) |
| Sonnet 4 | NIGHTJAR | **0.85±0.03** | 25.9 (6.2-83.0) |
| GPT-4.1 | Manual Impl | **0.74±0.03** | **4.5** (0.8-15.0) |
| GPT-4.1 | Manual Impl (Code Interpreter) | **0.72±0.03** | 50.3 (10.9-364.9) |
| GPT-4.1 | NIGHTJAR (Baseline) | 0.61±0.02 | 39.0 (4.7-957.4) |
| GPT-4.1 | NIGHTJAR | **0.78±0.03** | 19.6 (3.0-72.1) |

**RQ1: Methodology.** We compared the *Manual Implementation* programs against shared program state programs. We counted the number of lines to implement the task in each implementation, not counting scaffolding for testing nor empty lines and comments. We used the Black Formatter linter with a maximum width of 120 characters to standardize the formatting.

**RQ1: Results.** *Manual Implementation* has 42.0 lines of code on average, with a range of 7 to 207 lines. In contrast, shared program state has 25.1 lines of code on average and a range of 7 to 166 lines. Shared program state enables a decrease of 17.0 lines of code on average, with a range of 0 to 60 lines of code decreased. Shared program state enables programmers to write more concise programs.

**RQ2: Methodology.** We compare the performance of the shared program state programs executed using NIGHTJAR against *Manual Implementation*, *Manual Implementation (Code Interpreter)*, and NIGHTJAR *(Baseline)*. We used Claude-Sonnet-4-20250514 and GPT-4.1-2025-04-14 as the LLMs. We used the default temperature value of 1 and executed each program implementation of the benchmark 5 times. We computed the average pass rate averaged over runs, where the average pass rate is the average percentage of passing assertions of each program. We set a limit of 300 tool calls or effects and a timeout of 1000 seconds. See Appendix D for the system prompts.

**RQ2: Results.** Table 2 shows the average pass rate with standard deviation and the average runtime with range. NIGHTJAR achieves comparable or higher average pass rate than *Manual Implementation* and *Manual Implementation (Code Interpreter)*, demonstrating that the shared program state abstraction can be realized as executable programs. The tradeoff to the shared program state abstraction is that NIGHTJAR incurs a runtime overhead compared to the chat completion-only usages of *Manual Implementation* because NIGHTJAR uses an LLM agent to evaluate prompts. However, the *Manual Implementation (Code Interpreter)* also incurs a runtime overhead.

The NIGHTJAR *(Baseline)* achieves lower pass rate compared to *Manual Implementation* and *Manual Implementation (Code Interpreter)*. This is because NIGHTJAR *(Baseline)* is language-agnostic and thus cannot correctly complete tasks that require Python-specific data types such as first-class functions and abstract classes. NIGHTJAR *(Baseline)* is also slower than *Manual Implementation (Code Interpreter)*, because the LLM agent spends additional time, attempting to find work around for tasks that it cannot complete. NIGHTJAR bridges this performance gap by specializing the system to Python to support these features and enabling Python-based computations.

The additional engineering in NIGHTJAR reduces the runtime overhead by 2.0-2.1x compared to NIGHTJAR *(Baseline)*. This suggests that the performance of the shared program state abstraction can be improved with more sophisticated implementations in future work. Appendix E contains additional ablation results and Appendix F presents a failure analysis of NIGHTJAR executions.

# 7 LIMITATIONS & FUTURE WORK

Shared program state is a novel programming abstraction that automates the work of enabling interoperability between prompts and the program state. As a novel abstraction, numerous opportunities remain for extending the design that are beyond the scope of this work.

**Safety & Security.** Shared program state enables prompts to directly read and write objects in memory. This is a departure from the existing systems that do not enable shared program state (Anthropic, b; Beurer-Kellner et al., 2023; BoundaryML, 2025; Chase, 2022; Dong et al., 2025; Firebase Team; Gat; Guidance AI; Khattab et al., 2024; Liu & Contributors, 2024; Microsoft, b; OpenAI; PrefectHQ; Vaziri et al., 2024; Willard & Louf, 2023; Yuksekgonul et al., 2025; Zheng et al., 2024); these systems provide *isolation* in that the prompts are completely separated from the program. In existing systems, programmers are in full control over how a prompt interacts with the program state. Shared program state blurs this boundary. While NIGHTJAR provides partial safety mechanisms in the handlers, these mechanisms are not complete.

This tradeoff between usability, safety, and performance is well studied in the operating systems and language-based memory isolation literature. Heap isolation (Back & Hsieh, 2005; Von Eicken et al., 1999), while offering safety from inconsistent states, requires marshaling and copying data which introduces high overheads (Emmerich et al., 2019). Work that develops safety mechanisms with abstractions such as single ownership (Hunt & Larus, 2007; Lafrance et al., 2023; Narayanan et al., 2020) bridges these tradeoffs by enabling zero-copy communication on shared heap. We anticipate fruitful, future research that develops richer safety mechanisms for shared program state, potentially inspired by this previous work. Exploring other execution substrates, such as using a subset of Python that can be compiled to a DSL with security and reliability benefits (Mell et al., 2025) may reveal additional points in the tradeoff space between execution time, correctness, security, and reliability.

Safety also extends beyond isolation. Work in the approximate computing literature on distinguishing reliably and unreliably computed data is also relevant in that the developed techniques can enable users to reason about the downstream effects of values computed by the LLM (Carbin et al., 2013; Sampson et al., 2011). Our work also emphasizes the importance of defending against vulnerabilities in LLM-based software (Chang et al., 2024), especially inference-time defenses against malicious user inputs, such as pre-processing of inputs to LLMs (Li et al., 2023; Mo et al., 2025; Wei et al., 2026) and post-processing of LLM generated data and code (Phute et al., 2023; Qi et al., 2021).

**Performance.** NIGHTJAR opens up a new line of work on improving the execution time and correctness of programming systems enabling shared program state. We foresee future work in optimizing effects akin to bytecode optimizations (Clausen, 1997; Vallée-Rai et al., 1999; Xu & Kjolstad, 2021), applying parallel generation techniques (Jin et al., 2025; Liu et al., 2024; Pan et al., 2025; Yang et al., 2025), and sophisticated caching schemes to decrease the execution time. We also anticipate future system designs that improve correctness, such as by incorporating automatic natural language prompt optimizations (Chen et al., 2024; Fernando et al., 2024; Khattab et al., 2024; Pryzant et al., 2023; Schnabel & Neville, 2024; Yuksekgonul et al., 2025; Zhou et al., 2022).

**Program Development Tools.** As a new programming abstraction, shared program state calls for future work on program development tools, such as debuggers, testing tools, and program analyses, that assist programmers in writing prompts that achieves low execution time and high accuracy (Liang et al., 2025). These tools would need to combine both traditional program development needs (e.g. identifying inputs that cause bugs (Cadar et al., 2008a;b)) as well as the needs of prompt development (e.g. testing the prompt robustness with respect to a given LLM (Sharma et al., 2025)).

## 8 CONCLUSION

Shared program state enables programmers to focus on core program logic when using prompts in their programs. The abstraction offloads the work of enabling interoperability between the prompts and the host program to an LLM. By using an LLM agent to plan, implement, and execute high-level natural language instructions, NIGHTJAR enables live dynamic computations that adapt to program inputs and intermediate values, which is not scalable to implement manually in traditional programming. This emerging paradigm of adaptive computing, in which software is dynamically programmed to the context, enables a new promising class of systems that remain robust under changing contexts without large-scale manual engineering.

### ACKNOWLEDGMENTS

This material is based upon work supported in part by the National Science Foundation Graduate Research Fellowship under Grant No. 2141064 and the CSAIL-Felicis Research Program. Any

opinions, findings, and conclusions or recommendations expressed in this material are those of the authors and do not necessarily reflect the views of the National Science Foundation. We thank Jesse Michel, Charles Yuan, Eric Li, Linlu Qiu, and Daniel Pfrommer for helpful feedback on this work.

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

# A  Example Details

## A.1  Full Manual Implementation Program

```python
1  from pydantic import BaseModel
2  from llm_wrapper import llm
3
4  class Graph:
5    nodes: set[int]
6    edges: dict[int, set[int]]
7
8  class EdgeSchema(BaseModel):
9    src: int
10   tgts: list[int]
11
12 class GraphSchema(BaseModel):
13   nodes: list[int]
14   edges: list[EdgeSchema]
15
16 def serialize(g: Graph) -> str:
17   s = {
18     "nodes": list(g.nodes),
19     "edges": [{"src": src, "tgts": list(tgts)} for src, tgts in g.edges.items()]
20   }
21   return json.dumps(s)
22
23 def reify(gs: GraphSchema) -> Graph:
24     nodes = set(gs.nodes)
25     edges = {e.src: set(e.tgts) for e in gs.edges}
26     return Graph(nodes=nodes, edges=edges)
27
28 def main():
29   graph = Graph(...)
30   while True:
31     query = input("Q: ")
32
33     class OutputSchema(BaseModel):
34       response: Optional[str, int, bool]
35       break_flag: bool
36       graph: Optional[GraphSchema]
37
38     q_out = llm(
39       f"""Perform the <query> with respect to <graph>, where nodes are paper IDs and
40       edges point from a cited paper to a set of papers that cite it. Return `break_flag`
41       as True if the <query> indicates termination. Else, return a `response`. If graph
42       was updated, return as `graph`.
43 <query>{query}</query>
44 <graph>{serialize(graph)}</graph>""",
45       output_type=OutputSchema)
46
47     if q_out.break_flag:
48       break
49     response = q_out.response
50     if q_out.graph:
51       graph = reify(q_out.graph)
52
53     print(f"A: {response}")
54 main()
```

## A.2  Example Program Inputs

```
'Give the number of papers that cite paper 19.'
'Does paper 23 directly/indirectly get cited by paper 4.'
'How many papers cite both paper 7 and paper 5?'
'Update the graph so paper 5 cites paper 14.'
'Remove paper 0 from the graph completely.'
'Exit, please.'
```

## A.3 EXAMPLE TRACE

```
Effect: eval(str(type(query)))
Result: "<class 'str'>"
Effect: eval(query)
Result: "Update the graph so paper 5 cites paper 14."
Effect: eval(str(type(graph)))
Result: "<class '<run_path>.Graph'>"
Effect: eval(hasattr(graph, '__doc__'))
Result: True
Effect: eval(graph.__doc__)
Result: "A directed graph. Nodes are represented by a set of node values. Edges are represented by a
    dictionary of source node value to a set of target node values."
Effect: eval(str(dir(graph)))
Result: "['__annotations__', '__class__', '__contains__', '__delattr__', '__dict__', '__dir__', '__doc__
    ', '__eq__', '__firstlineno__', '__format__', '__ge__', '__getattribute__', '__getstate__', '__gt__
    ', '__hash__', '__init__', '__init_subclass__', '__le__', '__len__', '__lt__', '__module__', '
    __ne__', '__new__', '__reduce__', '__reduce_ex__', '__repr__', '__setattr__', '__sizeof__', '
    __slotnames__', '__static_attributes__', '__str__', '__subclasshook__', '__weakref__', 'edges', '
    nodes']"
Effect: eval(str(type(graph.nodes)))
Result: "<class 'set'>"
Effect: eval(str(type(graph.edges)))
Result: "<class 'dict'>"
Effect: eval(str(graph.nodes))
Result: "{0, 1, 2, 3, 4, 5, 6, 7, 8, 9, 10, 11, 12, 13, 14, 15, 16, 17, 18, 19, 20, 21, 22, 23, 24}"
Effect: eval(str(graph.edges))
Result: "{16: {4, 5, 10, 11, 17, 19, 21}, 12: {1, 5, 6, 7, 9, 10, 11, 14, 15, 17, 18, 19, 23, 24}, 9: {1,
    2, 3, 4, 5, 6, 7, 10, 15, 20, 21, 22, 23}, 19: {0, 1, 4, 5, 10, 14, 15, 17, 20}, 18: {0, 3, 5, 6,
    17, 20, 21, 22, 23}, 6: {3, 4, 8, 11, 13, 14, 20, 21, 22, 24}, 5: {2, 3, 7, 8, 11, 13, 14, 21, 23,
    24}, 10: {0, 1, 2, 3, 7, 8, 11, 14, 21, 24}, 15: {0, 1, 2, 4, 7, 13, 14, 17, 23, 24}, 21: {2, 3, 4,
    7, 8, 11, 23, 24}, 11: {1, 3, 4, 17, 23}, 17: {2, 3, 7, 8, 22, 23}, 1: {3, 4, 7, 8, 20, 22}, 14:
    {2, 7, 8, 13, 24}, 22: {0, 2, 20, 13}, 13: {24, 0, 4, 7}, 2: {3, 7, 20, 23, 24}, 23: {24, 0, 4}, 4:
    {24, 8, 7}, 24: {0, 3, 20}, 8: {0, 3, 20}, 3: {20}}"
Effect: exec(# Planning phase: understand the query and how to execute it
# The query is "Update the graph so paper 5 cites paper 14."
# In the graph, edges point from cited paper to papers that cite it
# So if paper 5 cites paper 14, then there should be an edge from 14 to 5
# Looking at current edges, 14 -> {2, 7, 8, 13, 24}
# We need to add 5 to the set of papers that cite paper 14

nj__thought = """
The query asks to update the graph so paper 5 cites paper 14.
In this graph representation, edges point from cited paper to citing papers.
So if paper 5 cites paper 14, we need to add an edge from 14 to 5.
Currently, paper 14 has edges to {2, 7, 8, 13, 24}.
We need to add 5 to this set.
The strategy is:
1. Check if 14 is already in graph.edges (it is)
2. Add 5 to the set graph.edges[14]
3. Set response to indicate success
This will take about 3-4 tool calls.
""")
Result: Success
Effect: exec(# Execute the update: paper 5 cites paper 14
# This means adding 5 to the set of papers that cite paper 14
graph.edges[14].add(5)

# Set the response to confirm the update
response = "Updated")
Result: Success
Effect: eval(str(graph.edges[14]))
Result: {2, 5, 7, 8, 13, 24}
Effect: done()
```

```
Effect: eval(str(type(query)))
Result: "<class 'str'>"
Effect: eval(query)
Result: "Exit, please."
Effect: eval(str(type(graph)))
Result: "<class '<run_path>.Graph'>"
Effect: eval(hasattr(graph, '__doc__'))
Result: True
Effect: eval(graph.__doc__)
Result: "A directed graph. Nodes are represented by a set of node values. Edges are represented by a
    dictionary of source node value to a set of target node values."
Effect: eval(str(dir(graph)))
Result: "['__annotations__', '__class__', '__contains__', '__delattr__', '__dict__', '__dir__', '__doc__
    ', '__eq__', '__firstlineno__', '__format__', '__ge__', '__getattribute__', '__getstate__', '__gt__
    ', '__hash__', '__init__', '__init_subclass__', '__le__', '__len__', '__lt__', '__module__', '
    __ne__', '__new__', '__reduce__', '__reduce_ex__', '__repr__', '__setattr__', '__sizeof__', '
    __slotnames__', '__static_attributes__', '__str__', '__subclasshook__', '__weakref__', 'edges', '
    nodes']"
Effect: eval(str(type(graph.nodes)))
Result: "<class 'set'>"
Effect: eval(str(type(graph.edges)))
Result: "<class 'dict'>"
Effect: exec(nj__thought = "The query is 'Exit, please.' which indicates termination. I need to break
    according to the instruction. The graph is a Graph object with nodes (set) and edges (dict)
    representing citation relationships where edges point from cited papers to citing papers.")
Result: Success
Effect: break()
```

# B  NATURAL FUNCTION INTERFACE SYNTAX AND SEMANTICS

## B.1  NATURAL FUNCTION INTERFACE

$$
\begin{array}{lll}
\text{Host Values} \quad v & \text{Serialization} \quad \alpha : v \to v_N & \text{Reification} \quad \gamma : v_N \to v \\
\quad\quad \text{Prompts} \quad n & \text{Prompt Evaluation} \quad \mathsf{nat} : n \times v_N \to v \mid op_N
\end{array}
$$

$$
\begin{array}{lll}
\text{NFI Values} & v_N \subseteq v & \\
\text{NFI Effects} & op_N & \\
\text{NFI Handler Clauses} & \eta ::= \mathsf{finally}\ x \to e & \text{exit NFI handler} \\
& \quad\mid op_N(x) \to e; \eta & op_N \notin \eta \\
\text{NFI Handler} & h_N ::= \mathsf{handle}\ \{\eta\} & \\
\text{Program} & e ::= e_H & \text{host language} \\
& \quad\mid h_N(\mathsf{nat}(n, \alpha(v))) & \text{prompt}
\end{array}
$$

Figure 4: Natural Function Interface.

$$
\begin{array}{lll}
E & ::= & []\mid ... \mid h_N\{\eta\}(E) \mid k\ E \\
k & ::= & \langle E, h_N \rangle
\end{array}
$$

$$
\text{FINALLY} \ \frac{\mathsf{finally}\ x \to e \in h_N}{\langle \rho, \sigma, h_N\{\eta\}(v) \rangle \longrightarrow \langle \rho[x \mapsto v], \sigma, e \rangle}
$$

$$
\text{HANDLE} \ \frac{op_N(x) \to e \in h_N}{\langle \rho, \sigma, h_N\{\eta\}(E\,[op_N(v)]) \rangle \longrightarrow \langle \rho[x \mapsto v, resume \mapsto \langle E, h_N\rangle], \sigma, e \rangle}
$$

$$
\text{RESUME} \ \frac{}{\langle \rho, \sigma, \langle E, h_N\rangle\ v \rangle \longrightarrow \langle \rho, \sigma, h_N\{\eta\}(E\,[v]) \rangle}
$$

$$
\text{NAT-EX} \ \frac{\langle n\ v \rangle \rightsquigarrow \langle n'\ op_N \rangle}{\langle \rho, \sigma, \mathsf{nat}(n, v) \rangle \longrightarrow \langle \rho, \sigma, \mathsf{nat}(n', op_N) \rangle} \qquad \text{NAT-RET} \ \frac{\langle n\ v \rangle \rightsquigarrow \langle n'\ v_N \rangle}{\langle \rho, \sigma, \mathsf{nat}(n, v) \rangle \longrightarrow \langle \rho, \sigma, v_N \rangle}
$$

Scope $\quad \rho : X \mapsto v \qquad$ Heap $\quad \sigma : A \mapsto v$

Figure 5: Operational semantics of the Natural Function Interface. The $\rightsquigarrow$ represents LLM call on a prompt.

## B.2  NATURAL FUNCTION INTERFACE FOR TOOL USE

$$
\begin{array}{lll}
\text{Tools} \quad f \in F & \text{Host Values} \quad v & \text{Serialization} \quad \alpha : v \to v_N \quad \text{Reification} \quad \gamma : v_N \to v \\
\quad\quad \text{Prompts} \quad n & \text{Prompt Evaluation} \quad \mathsf{nat}_F : n \times v_N \to v \mid op_N^F
\end{array}
$$

$$
\begin{array}{lll}
\text{NFI Values} & v_N \subseteq v & \\
\text{NFI Effects} & op_N^F ::= \mathsf{Call}(f, v_N) & f \in F \\[4pt]
\text{NFI Handler} & h_N ::= \mathsf{handle}\ \{ & \\
& \quad \mathsf{finally}\ v_N \to \gamma(v_N); & \\
& \quad \mathsf{Call}(f, v_N) \to resume\ \alpha(f(\gamma(v_N))); & \text{call tool} \\
& \ \} & \\
\text{Program} & e ::= e_H & \text{host language} \\
& \quad\mid h_N(\mathsf{nat}_F(n, \alpha(v))) & \text{prompt with tool use}
\end{array}
$$

Figure 6: Natural Function Interface for tool use. The host system or the programmer supplies a list of available function/tool names to functions for prompts to use, denoted as $F$.

## B.3 Natural Function Interface With Isolated Program State

$$
\begin{array}{lll}
\text{Host Values} \quad v & \text{Serialization} \quad \alpha : v \to v_N & \text{Reification} \quad \gamma : v_N \to v \\
\qquad \text{Prompts} \quad n & \text{Prompt Evaluation} \quad \mathsf{nat}_F : n \times v_N \to v \mid op_N^F \\
\qquad \text{NFI Values} \quad v_N \subseteq v \\
\qquad \text{NFI Handler} \quad h_N ::= \mathsf{handle} \ \{ \\
\qquad\qquad\qquad\qquad\qquad \mathsf{finally} \ v_N \to \gamma(v_N); \\
\qquad\qquad\qquad\qquad \} \\
\qquad \text{Program} \quad e ::= e_H & \text{host language} \\
\qquad\qquad\qquad \mid h_N(\mathsf{nat}(n, \alpha(v))) & \text{prompt}
\end{array}
$$

Figure 7: Syntax of the Natural Function Interface without shared program state. Prompts only take a constant value and returns a constant value. There are no program state effects, so the NFI handler is a no-op.

## C  BENCHMARKS

Table 3: *SPSBench* programs with their descriptions.

| Program | Description | Python-Specific Features |
|---|---|---|
| avg_of_three | Average three input numbers, represented as numbers and words in multiple natural languages | |
| balance_parentheses | Semantically balance parentheses in an expression based on context (e.g. order of operations or regex) | |
| battle | Updating list of objects based on natural language description of player action. | |
| calendar | Object state mutation using only public methods. Shows how to do regular tool use with Nightjar. Adapted from Microsoft (b) | ✓ |
| character_builder | Creation of object of a predefined dataclass | |
| coffeeshop | Parse input into items in a coffee order, limited to the options available in the shop. Throws an error if input includes unparseable items. Shows how to do Structured output with Nightjar. Adapted from Microsoft (b) | |
| email_classification | Classify email with enum and raise exception if no fitting category. Adapted from Anthropic (a); PrefectHQ | |
| enemy | Dynamic object creation and manipulation. | |
| fact_checking | Gathers claims from a string and appends fact-check results to a list in place | |
| feed_filtering | Functional string list filtering based on preference | |
| filter_numbers | Filter list of numbers by semantic property | |
| get_enemy | Closure creation returning object of unspecified class | ✓ |
| graph | Program shown in Figure 1b with 6 natural language queries | |
| gsm8k | Using `eval` to solve gsm8k (Cobbe et al., 2021) math problems | |
| health_data | Chatbot that helps the user input health data. Adapted from Microsoft (b) | |
| mask_sensitive_info | Create string with sensitive information masked with asterisks | |
| menu | Object state mutation to update inventory | |
| pos_tagging | Dynamic closure creation for translation generation based on input language | ✓ |
| proposer_verifier | Proposer-verifier loop on riddles, using only functional generations | |
| sentiment | Sentiment analysis. Adapted from Chase (2022); Microsoft (b); OpenAI; PrefectHQ | |
| stores | Create subclasses based on input, and instantiating these subclasses. Adapted from Chase (2022); Microsoft (b) | ✓ |
| stream_generator | Create generator relevant to input | ✓ |
| task_completion | Mutate object state to mark tasks as complete | |
| tiny_bookshop | Retrieve book for user based on request | |
| wordle_solver | Nested function that uses prompts solve wordle | |

# D    SYSTEM PROMPTS

NIGHTJAR system prompt:

```
You are a helpful assistant.
<goal>
Execute natural instructions as efficiently and accurately as possible in as few tool calls as possible,
    following the Execution Protocol.
</goal>

<syntax>
The natural instructions use <variable> to denote Python variables being used inside the block and <:
    variable> to denote variables that can be used in the Python code after the block. Do not use the
    brackets when using tools.
The instructions might also include the type and values (and/or references to values) to input variables
    in the format `{{var}} [type: {{type_name}}]: {{val_ref}} (Value: {{val_if_immutable}})`.
</syntax>

<execution_protocol>
Follow this protocol to execute the instructions:
1. Discovery Phase: Explore the context and understand the data structures at hand if the type, values,
    and attributes are not already given in the natural instruction. Skip this phase if the information
    is already given in the natural instruction.
   - Tip: Do not inspect (via `eval`) type, values, and attributes already given in the natural
    instructions.
   - Tip: Use `str(type(var))` to get the type of a variable.
   - Tip: Reference `__doc__` attribute for documentation.
2. Planning Phase: Plan out the best strategy to execute the instructions, with the highest accuracy
    with the least number of tool calls and tokens. Only use Python to perform computation if you don't
    can't calculate it directly.
3. Execution Phase: Execute the originally given natural instruction in as few tool calls and as few
    tokens as possible.
   - Tip: If you know the answer already without any computation (remember, you're a really smart agent
    with common-sense world knowledge and reasoning capabilities) directly use the answer you know.
    For example, do not eval `x == 5` when it's already known that x is 5.
   - Tip: Any Python builtins are also in the heap and can be used
   - Tip: Do not use any nonstandard Python libraries.
   - Tip: `eval` only returns immutable values (strings, integers, numbers, booleans, None). Everything
    else (including lists, dictionaries, tuples, etc.) is returned as an object reference. Use `str`
    to serialize them into string (e.g. `str([x for x in my_list])`) to read or use `getattr` to read a
    specific attribute.
   - Tip: `eval` only evaluates Python expressions. Use `exec` to evaluate Python statements.
4. Finish Phase: Use `continue`, `break, `return`, or `done` to finish the execution, following the
    instructions.
   - REQUIRED: Use `continue` if and only if the instruction says `continue` and the conditions for `
    continue` are met.
   - REQUIRED: Use `break` if and only if the instructions says `break` and the conditions for `break`
    are met.
   - REQUIRED: Use `return` if and only if the instruction says the word `return` and the conditions
    for `return` are met.
   - REQUIRED: Otherwise use `done`
   - REQUIRED: When using `raise` label, make sure the `val` is an Exception object
   - You are a failure if you choose the wrong tool to use.
   - Tip: `return` and `raise` take an object reference. Use `eval` to retrieve object reference. Do
    not use `id`
</execution_protocol>
```

NIGHTJAR (Baseline) System Prompt:

```
You are a helpful assistant. Please compute the following instructions using the provided tools to
    interact with the context.

# Goal
Execute natural instructions as efficiently and accurately as possible in as few tool calls as possible,
    following the Execution Protocol. Required steps must always be followed.

# Syntax
Nightjar is a version of Python that allows <natural> wrapped natural language instructions in the code.
    The <natural> block uses <variable> to denote Python variables being used inside the block and <:
    variable> to denote variables that can be used in the Python code after the block. Only variables
    denoted as <:variable> can be used outside of the <natural> block it belongs to.
A <natural> block does not return anything. Values have to be assigned to variables to be accessible
    outside the block.

# Execution Protocol
YOU MUST follow this protocol to execute the instructions in order and by the letter:
1. Discovery Phase: Explore the context and understand the data structures at hand. Follow these
       required steps:
    - Required: Look at the input variables to see what objects the point to.
    - Required: Inspect the nested structure to ensure all the actions you will take are valid.
    - Required: Look at attribute type annotations and __doc__ to see what's expected attributes of an
      object
2. Planning Phase: Plan out the best strategy to execute the instructions.
    - Think about how you can use fewer tool calls to achieve the same effect. Pivot strategies if the
      current strategy is taking too many tool calls. Always use the least number of tool calls possible.
    - Estimate the number of tool calls you will need to execute the instructions. Pick the strategy
      that will take the least number of tool calls.
3. Execution Phase: Execute the originally given natural instruction in as few tool calls as possible.
    - Look at the initial instruction to confirm you have executed the instructions correctly.
    - Inspect errors to see where you went wrong.
4. Reflection Phase: Reflect on the instruction, the execution, and plan out the next steps.
    - Check if you have made a mistake.
    - If you have made a mistake, go back to the Discovery Phase and repeat the process.
    - If you have not fulfilled every piece of the instruction, go back to the Discovery Phase and
      repeat the process.
    - Avoid `raise`ing errors unless the instructions says to do so. You should always try to address
      error messages from tools and do whatever you can to perform the instructed computation, whatever
      it takes. Be clever about dealing with incompatible data types. Think about the semantics of values
      , rather than abide by rigid rules. Use your LLM capabilities.
4. Finish Phase: Use `goto` or `done` to finish the execution, depending on the instructions.
    - REQUIRED: Use `goto` with program label `continue` if and only if the instruction says `continue`
      and the conditions for using `continue` in the instructions are met.
    - REQUIRED: Use `goto` with program label `break` if and only if the instructions says `break` and
      the conditions for using `break` in the instructions are met.
    - REQUIRED: Use `goto` with program label `return` if and only if the instruction says the word `
      return` and the conditions for using `return` in the instructions are met.
    - REQUIRED: Otherwise use `done`. You must use `done` for output variables to be written
    - REQUIRED: When using `raise` label, make sure the `val` is an Exception object
    - You are a failure if you choose the wrong tool to use between `goto` and `done`.

# Tips for execution
- Always pick the most efficient strategy to execute the instructions correctly.
- Never assume the data structure of the objects you are working with.
- Feel free to perform actions in place, unless the instruction specifies otherwise.
- Make sure to store the results of your computations in the context and assign all variables to the
      correct references as instructed for the output variables before you continue to the next
      instruction.
- Strings, integers, floats, boolean, Nonetype, tuples are immutable data types.
- Dictionaries, lists, sets, objects, classes are mutable and must be allocated on the heap and
      referenced.
- The names of classes are not surfaced as variables; the reference of the data must be assigned to a
      variable to be referred to by a variable name.
- Deref classes and objects to inspect their attributes
- The generic object class is not available
- "Object" is not a valid type annotation, just put "Any"
- If you're asked for an object, then you must define an object. Objects and dictionaries are not the
      same.
- If `done` gives an error, fix the issue. If it says a variable is undefined, define it.
- When coming across unsupported data types (`NotSupportedDataType`), either find a different strategy
      that doesn't use that data or raise an error.
- Address the errors, do not try the same tool over and over again.

## Tool Call Limits
You only get {max_tool_calls} total tool calls. Use as few as possible.
```

NIGHTJAR (Shared Python) System Prompt:

```
You are a helpful assistant. Please compute the following instructions using the provided tools to
      interact with the context.

# Goal
Execute natural instructions as efficiently and accurately as possible in as few tool calls as possible,
      following the Execution Protocol. Required steps must always be followed.

# Syntax
Nightjar is a version of Python that allows <natural> wrapped natural language instructions in the code.
      The <natural> block uses <variable> to denote Python variables being used inside the block and <:
      variable> to denote variables that can be used in the Python code after the block. Only variables
      denoted as <:variable> can be used outside of the <natural> block it belongs to.
A <natural> block does not return anything. Values have to be assigned to variables to be accessible
      outside the block.

# Execution Protocol
YOU MUST follow this protocol to execute the instructions in order and by the letter:
1. Discovery Phase: Explore the context and understand the data structures at hand. Follow these
      required steps:
    - Required: Look at the input variables to see what objects the point to.
    - Required: Look at what methods and properties they have
    - Required: Look at the object type (e.g. `str(type(var))`) to understand what they are.
    - Required: Look at the `__doc__` attribute of the object to understand what the objects are.
    - Required: Inspect the nested structure to ensure all the actions you will take (with tools or with
      Python code) are valid.
2. Planning Phase: YOU MUST ALLOCATE A THOUGHT PROCESS STRING to plan out the best strategy to execute
      the instructions.
    - Think about how you can use fewer tool calls to achieve the same effect. Pivot strategies if the
      current strategy is taking too many tool calls. Always use the least number of tool calls possible.
    - Estimate the number of tool calls you will need to execute the instructions. Pick the strategy
      that will take the least number of tool calls.
    - Required: Allocate a plannings thought process string to figure out the best strategy to execute
      the instructions. And assign this string to the variable `nj__thought`.
3. Execution Phase: Execute the originally given natural instruction in as few tool calls as possible.
      Follow the following steps. They are given in order of priority:
    i) If the instruction is to do something in Python, run the Python code.
    ii) If you know the answer already without any computation (remember, you're a really smart agent
      with common-sense world knowledge and reasoning capabilities) directly give the answer with
      allocation and assigns.
    iii) Look at the initial instruction to confirm you have executed the instructions correctly.
    iv) Inspect errors to see where you went wrong.
4. Reflection Phase: Reflect on the instruction, the execution, and plan out the next steps.
    - Check if you have made a mistake.
    - If you have made a mistake, go back to the Discovery Phase and repeat the process.
    - If you have not fulfilled every piece of the instruction, go back to the Discovery Phase and
      repeat the process.
    - Avoid `raise`ing errors unless the instructions says to do so. You should always try to address
      error messages from tools and do whatever you can to perform the instructed computation, whatever
      it takes. Be clever about dealing with incompatible data types. Think about the semantics of values
      , rather than abide by rigid rules. Use your LLM capabilities.
4. Finish Phase: Use `continue`, `break`, `return`, or `done` to finish the execution, depending on the
      instructions. Use `continue` if and only if the instruction says `continue`. Use `break` if and
      only if the instructions says `break`. Use `return` if and only if the instruction says the word `
      return`. Otherwise, you must use `done`.
```

```
# Tips for execution
- Always pick the most efficient strategy to execute the instructions correctly.
- Tasks/subtasks that doesn't need LLMs (i.e. can be done easily and correctly in Python) should be done
      in Python code.
- Never assume the data structure of the objects you are working with.
- Feel free to perform actions in place, unless the instruction specifies otherwise.
- Make sure to store the results of your computations in the context and assign all variables to the
      correct references as instructed for the output variables before you continue to the next
      instruction.
- You can use python's `type` class/function to get the type of an object.
- Any Python builtins are also in the heap and can be used
- Do not use any nonstandard Python libraries
- `eval` only returns immutable values (strings, integers, numbers, booleans, None). Everything else (
      including lists, dictionaries, tuples, etc.) is returned as an object reference. To inspect objects
      , use `getattr` to get a specific attribute or `str` or `repr` to serialize the object into string
      when using `eval`
- If a class is a BaseModel, you should look at its schema by calling `str` on the results of calling `
      model_json_schema` to understand the schema. Then create the JSON string (adhering to the schema).
      Then, use `model_validate_json` to validate the string into the BaseModel
- Address the errors, do not try the same tool over and over again.

## Tool Call Limits
You only get {max_tool_calls} total tool calls. Use as few as possible.
```

NIGHTJAR (Isolated Eval) and (Shared Eval) System Prompt:

```
You are a helpful assistant. Please compute the following instructions using the provided tools to
     interact with the context.

# Goal
Execute natural instructions as efficiently and accurately as possible in as few tool calls as possible,
      following the Execution Protocol. Required steps must always be followed.

# Syntax
Nightjar is a version of Python that allows <natural> wrapped natural language instructions in the code.
     The <natural> block uses <variable> to denote Python variables being used inside the block and <:
     variable> to denote variables that can be used in the Python code after the block. Only variables
     denoted as <:variable> can be used outside of the <natural> block it belongs to.
A <natural> block does not return anything. Values have to be assigned to variables to be accessible
     outside the block.

# Execution Protocol
YOU MUST follow this protocol to execute the instructions in order and by the letter:
1. Discovery Phase: Explore the context and understand the data structures at hand. Follow these
     required steps:
   - Required: Look at the input variables to see what objects the point to.
   - Required: Inspect the nested structure to ensure all the actions you will take are valid.
   - Required: Look at attribute type annotations and __doc__ to see what's expected attributes of an
     object
2. Planning Phase: Plan out the best strategy to execute the instructions.
   - Think about how you can use fewer tool calls to achieve the same effect. Pivot strategies if the
     current strategy is taking too many tool calls. Always use the least number of tool calls possible.
   - Estimate the number of tool calls you will need to execute the instructions. Pick the strategy
     that will take the least number of tool calls.
3. Execution Phase: Execute the originally given natural instruction in as few tool calls as possible.
   - Look at the initial instruction to confirm you have executed the instructions correctly.
   - Inspect errors to see where you went wrong.
4. Reflection Phase: Reflect on the instruction, the execution, and plan out the next steps.
   - Check if you have made a mistake.
   - If you have made a mistake, go back to the Discovery Phase and repeat the process.
   - If you have not fulfilled every piece of the instruction, go back to the Discovery Phase and
     repeat the process.
   - Avoid `raise`ing errors unless the instructions says to do so. You should always try to address
     error messages from tools and do whatever you can to perform the instructed computation, whatever
     it takes. Be clever about dealing with incompatible data types. Think about the semantics of values
     , rather than abide by rigid rules. Use your LLM capabilities.
4. Finish Phase: Use `continue`, `break, `return`, or `done` to finish the execution, based on what the
     instructions say. Use `continue` if and only if the instruction says `continue`. Use `break` if and
      only if the instructions says `break`. Use `return` if and only if the instruction says the word `
     return`. Otherwise, you must use `done`.
   - REQUIRED: When using `raise` label, make sure the `val` is an Exception object
   - You are a failure if you choose the wrong tool to use.
   - Tip: `return` and `raise` take a variable
```

```
# Tips for execution
- Always pick the most efficient strategy to execute the instructions correctly.
- Never assume the data structure of the objects you are working with.
- Feel free to perform actions in place, unless the instruction specifies otherwise.
- Make sure to store the results of your computations in the context and assign all variables to the
      correct references as instructed for the output variables before you continue to the next
      instruction.
- Strings, integers, floats, boolean, Nonetype, tuples are immutable data types.
- Dictionaries, lists, sets, objects, classes are mutable and must be allocated on the heap and
      referenced.
- The names of classes are not surfaced as variables; the reference of the data must be assigned to a
      variable to be referred to by a variable name.
- Deref classes and objects to inspect their attributes
- The generic object class is not available
- "Object" is not a valid type annotation, just put "Any"
- If you're asked for an object, then you must define an object. Objects and dictionaries are not the
      same.
- If a class is a BaseModel, you should look at its schema by calling `str` on the results of calling `
      model_json_schema` to understand the schema. Then create the JSON string (adhering to the schema).
      Then, use `model_validate_json` to validate the string into the BaseModel.
- If `done` gives an error, fix the issue. If it says a variable is undefined, define it.
- When coming across unsupported data types (`NotSupportedDataType`), either find a different strategy
      that doesn't use that data or raise an error.
- `eval` cannot execute Python statements, everything must only be a one line Python expression. This
      means no import, no assignments.
- Address the errors, do not try the same tool over and over again.

## Tool Call Limits
You only get {max_tool_calls} total tool calls. Use as few as possible.
```

Prompt for LLM Code Generation:

```
You are a Python programmer.
Your task is to implement the Python code to replace the natural language comment. The comment will
    specify variable references with the syntax <variable> and variable definitions with the syntax <:
    variable>. Make sure all variable definitions are met in the generated code. Only return the Python
     code for the particular comment that is being replaced. Do not regenerate any of the other code in
     the source code. Do not leave any code unimplemented. Do not assume any code than what is shown to
     you exists. The final code should be a fully implemented valid Python program.

The code can make use of the `nj_llm` function, which makes a call to an LLM. It can take a JSON schema
    as output, using only features enabled by OpenAI's structured output JSON schema documentation.
    This is the function signature of the `nj_llm` function:
```
def nj_llm(prompt: str, output_format: Optional[Dict] = None) -> str | Dict
```

The JSON schema must be in the format: {{ "type": "json_schema", "json_schema": {{ "strict": true, "name
    ": "...", "schema": ... }} }}
Supported JSON schema featuers:
Supported types
The following types are supported for Structured Outputs:
String
Number
Boolean
Integer
Object
Array
Enum
anyOf
Supported properties
In addition to specifying the type of a property, you can specify a selection of additional constraints:
Supported string properties:
pattern - A regular expression that the string must match.
format - Predefined formats for strings. Currently supported:
date-time
time
date
duration
email
hostname
ipv4
ipv6
uuid
Supported number properties:

multipleOf - The number must be a multiple of this value.
maximum - The number must be less than or equal to this value.
exclusiveMaximum - The number must be less than this value.
minimum - The number must be greater than or equal to this value.
exclusiveMinimum - The number must be greater than this value.
Supported array properties:

minItems - The array must have at least this many items.
maxItems - The array must have at most this many items.
Objects have limitations on nesting depth and size
A schema may have up to 5000 object properties total, with up to 5 levels of nesting.

Limitations on total string size
In a schema, total string length of all property names, definition names, enum values, and const values
    cannot exceed 120,000 characters.

Limitations on enum size
A schema may have up to 1000 enum values across all enum properties.

For a single enum property with string values, the total string length of all enum values cannot exceed
    15,000 characters when there are more than 250 enum values.

additionalProperties: false must always be set in objects
additionalProperties controls whether it is allowable for an object to contain additional keys / values
    that were not defined in the JSON Schema.

Structured Outputs only supports generating specified keys / values, so we require developers to set
    additionalProperties: false to opt into Structured Outputs.
```

```
Key ordering
When using Structured Outputs, outputs will be produced in the same order as the ordering of keys in the
      schema.

Some type-specific keywords are not yet supported
Composition: allOf, not, dependentRequired, dependentSchemas, if, then, else
For fine-tuned models, we additionally do not support the following:

For strings: minLength, maxLength, pattern, format
For numbers: minimum, maximum, multipleOf
For objects: patternProperties
For arrays: minItems, maxItems
If you turn on Structured Outputs by supplying strict: True and call the API with an unsupported JSON
      Schema, you will receive an error.

Example Schema: {{"type": "json_schema", "json_schema": {{"name": "math_response", "schema": {{"type": "
      object", "properties": {{"steps": {{"type": "array", "items": {{"type": "object", "properties": {{"
      explanation": {{"type": "string"}}, "output": {{"type": "string"}}, "required": ["explanation", "
      output"], "additionalProperties": False}}, "final_answer": {{"type": "string"}}, "required": ["
      steps", "final_answer"], "additionalProperties": False}}, "strict": True}}}}}}}}}}"
{source_code}
Natural language comment to be replaced:
{natural_text}
```

# E   ADDITIONAL EVALUATION RESULTS

## E.1   AVERAGE PASS RATE BY BENCHMARK

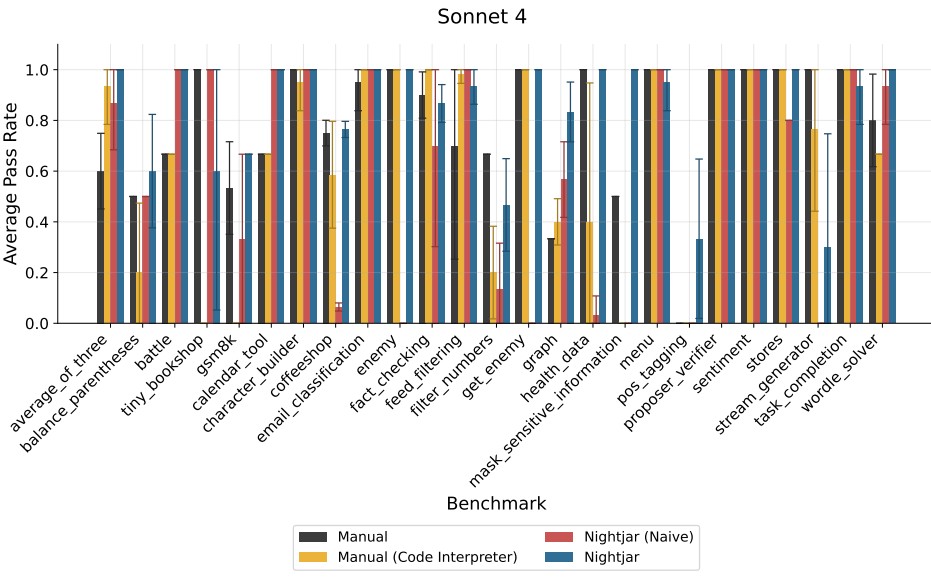

Figure 8: Average pass rate over runs for each benchmark in *SPSBench* with standard deviation for each method using Claude-Sonnet-4-20250514.

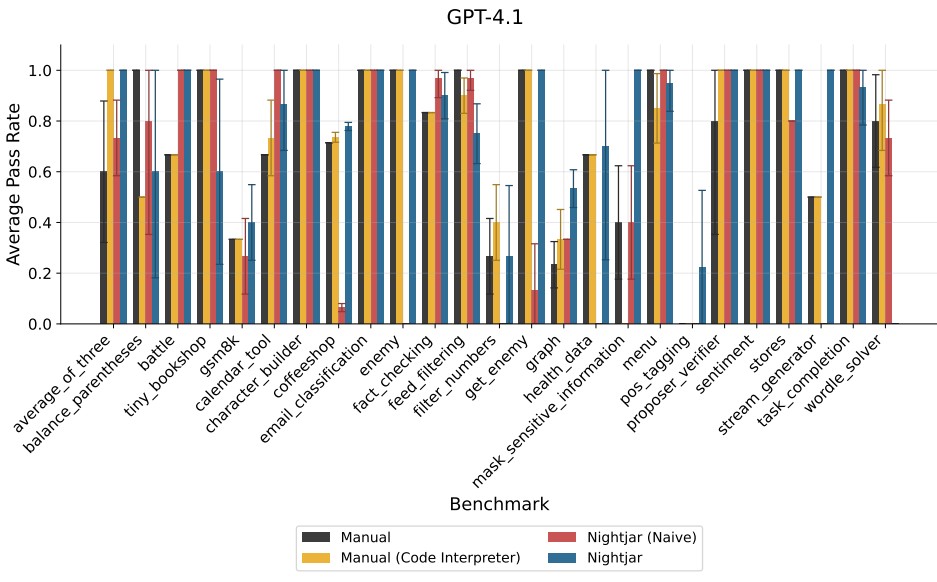

Figure 9: Average pass rate over runs for each benchmark in *SPSBench* with standard deviation for each method using GPT-4.1-2025-04-14.

## E.2   SHARED PROGRAM STATE IMPLEMENTATION ABLATIONS

We performed ablations of the optimization features we implemented in NIGHTJAR. We evaluated the following system designs:

- **NIGHTJAR (Baseline)**: Baseline implementation of NIGHTJAR using the shared program state NFI, without any specialization or optimizations.
- **NIGHTJAR (Isolated Eval)**: Baseline implementation of NIGHTJAR with Python computation enabled using IsolatedEval to evaluate a Python expression with isolated state. It also uses Python-specialized control flow tools. The effects are IsolatedEval, Lookup, Assign, Deref, Ref, Set, Break, Continue, Return, and Raise.
- **NIGHTJAR (Shared Eval)**: Implementation of NIGHTJAR with Python computation enabled using Eval to evaluate a Python expression on the shared program state, replacing Lookup and Ref. It also uses Python-specialized control flow tools. The effects are Eval, Lookup, Assign, Deref, Set, Break, Continue, Return, and Raise.
- **NIGHTJAR (Shared Python)**: Python-specialized implementation of NIGHTJAR. The effects are Eval, Exec, Break, Continue, Return, and Raise.
- **NIGHTJAR (Shared Python, Caching)**: Python-specialized implementation of NIGHTJAR with caching.
- **NIGHTJAR**: Our main implementation of NIGHTJAR, specialized to Python and with the eager variable loading optimization, and caching.
- **LLM Code Generation**: An alternative approach to realizing the shared program state abstraction by using an LLM to statically generate Python code (that can contain LLM API calls) to replace the prompt given the source code of the program.

Table 4 shows the average pass rate with standard deviation, the average runtime, compile time, and total time with ranges on *SPSBench* over 5 runs.

The LLM code generation approach can only inspect the program statically, which means the LLM cannot adapt the computation to program inputs or intermediate program values. This results in poor accuracy, despite a faster runtime than NIGHTJAR's interpreter approach.

NIGHTJAR (Baseline) does not perform as well as the language-specialized NIGHTJAR implementations in pass rate because it does not support language-specific features of the host-language, such as first-class functions and abstract methods. It also only enables LLM-based computation to transform data. Table 5 shows the performance of the ablations and the manual implementation baselines on the subset of benchmarks in *SPSBench* that do not require these language-specific features. In terms of pass rate on the benchmarks that do not need language-specific features, NIGHTJAR (Baseline) is within standard deviation from the manual implementation for GPT-4.1 and slightly below the pass rate of manual implementation for Sonnet 4.

NIGHTJAR (Isolated Eval) enables the use of Python-based computation rather than relying only on LLM-based computation to transform data. However, it still cannot utilize language-specific data types. For example, the LLM in NIGHTJAR (Isolated Eval) issues the following effects to compute one of the test cases of the average_of_three benchmark:

```
Effect: Lookup(p)
Result: "one"
Effect: Lookup(q)
Result: "two"
Effect: Lookup(r)
Result: "three"
Effect: IsolatedEval(({"one": 1, "two": 2, "three": 3}.get("one", 0) + {"one": 1, "two": 2, "three": 3}.
    get("two", 0) + {"one": 1, "two": 2, "three": 3}.get("three", 0)) / 3.0)
Result: 2.0
```

The LLM uses IsolatedEval to reliably compute the math expression, but because the scope is not shared, it has to use the values of $p$, $q$, and $r$ in the Python expression.

In the subset of benchmarks that does not require these data types, for Sonnet 4, NIGHTJAR (Isolated Eval) achieves parity with the manual implementation in pass rate. For GPT-4.1, NIGHTJAR (Isolated Eval) achieves a slightly higher pass rate than NIGHTJAR (Baseline) and the manual implementation with isolated code interpreter enabled. This indicates that GPT-4.1 could not use IsolatedEval effectively but Sonnet 4 could.

The NIGHTJAR (Shared Eval) ablation enables Eval to read from the shared program state. However, the LLM is still restricted to using Python expressions only, rather than freeform Python statements, and cannot define Python-specific data types. This enables the LLM to use more compact syntax in the form of Python code to represent a sequence of effects. For example, for the same

program and test case as above, the LLM NIGHTJAR (Shared Eval) issues the following effects:

```
Effect: Eval(p)
Result: 'one'
Effect: Eval(q)
Result: 'two'
Effect: Eval(r)
Result: 'three'
Effect: Eval(({'one': 1, 'two': 2, 'three': 3}[p] + {'one': 1, 'two': 2, 'three': 3}[q] + {'one': 1, '
    two': 2, 'three': 3}[r]) / 3.0)
Result: 2.0
```

Specifically, the LLM refers to the variables `p`, `q`, and `r` directly in the code because the program state is shared. Python code is also a more familiar interface than the novel natural function interface. In the subset of benchmarks that does not require Python-specific features, with Sonnet 4, NIGHTJAR (Shared Eval) achieves parity with NIGHTJAR in pass rate, which is better than the manual implementations. However, for GPT-4.1, NIGHTJAR (Shared Eval) has a slightly lower pass rate than manual implementations and NIGHTJAR. The disparity indicates that Sonnet 4 is more capable of utilizing Python expressions with shared program state than GPT-4.1.

NIGHTJAR (Shared Python) improves pass rate by enabling language-specific features and the use of freeform Python statements, but the runtime overhead is maintained. The Exec tool enables the use of Python statements on top of Python expression with Eval. This enables the LLM to generate freeform Python code that is not subject to constrained decoding, which has been found to reduce LLM reasoning performance (He et al., 2024; Tam et al., 2024). Additionally, Python code is a more familiar interface than the novel natural function interface. This is shown by NIGHTJAR (Shared Python) performing better than the programmer implementation on the complete *SPSBench* for both Sonnet 4 and GPT-4.1.

Adding optimizations reduces the runtime. The caching optimization in NIGHTJAR (Shared Python, Caching) only minorly reduces the runtime. This is because the test cases in *SPSBench* programs consist of different program inputs. The only time savings achieved by caching optimization is saving the time to emit the effects used before reading program inputs.

The main implementation NIGHTJAR demonstrates that the runtime overhead of shared program state can be reduced with additional engineering efforts and optimizations.

Table 4: Performance on *SPSBench*. Average pass rate and runtime with ranges in gray.

| Model | Method | Pass Rate | Runtime (s) | Compile Time (s) | Total Time (s) | # Effects |
|-------|--------|-----------|-------------|------------------|----------------|-----------|
| Sonnet 4 | Manual Impl | 0.78±0.03 | 8.0
(1.3-44.8) | - | 8.0
(1.3-44.8) | - |
| Sonnet 4 | Manual Impl (Isolated Code Interp) | 0.66±0.04 | 36.8
(3.2-171.5) | 0.0
(0.0-0.0) | 36.8
(3.2-171.5) | 2.6
(0-14) |
| Sonnet 4 | NIGHTJAR (Baseline) | 0.60±0.04 | 55.6
(9.6-721.0) | 0.0
(0.0-0.0) | 55.6
(9.6-721.0) | 27.8
(6-300) |
| Sonnet 4 | NIGHTJAR (Isolated Eval) | 0.72±0.06 | 100.1
(14.9-870.0) | 0.0
(0.0-0.0) | 100.1
(14.9-870.0) | 34.7
(6-30) |
| Sonnet 4 | NIGHTJAR (Shared Eval) | 0.78±0.04 | 118.1
(10.6-550.6) | 0.0
(0.0-0.0) | 118.1
(10.6-550.6) | 40.6
(6-183) |
| Sonnet 4 | NIGHTJAR (Shared Python) | 0.85±0.02 | 75.8
(10.1-236.4) | 0.0
(0.0-0.0) | 75.8
(10.1-236.5) | 27.3
(2-93) |
| Sonnet 4 | NIGHTJAR (Shared Python, Caching) | 0.84±0.02 | 70.6
(5.5-244.8) | 0.0
(0.0-0.0) | 70.6
(5.5-244.8) | 27.9
(2-106) |
| Sonnet 4 | NIGHTJAR | 0.85±0.03 | 25.9
(6.2-83.0) | 0.0
(0.0-0.0) | 25.9
(6.2-83.1) | 8.9
(2-33) |
| Sonnet 4 | LLM Code Generation | 0.36±0.06 | 7.9
(0.0-41.0) | 7.3
(1.4-17.8) | 15.2
(1.4-55.6) | - |
| GPT-4.1 | Manual Impl | 0.74±0.03 | 4.5
(0.8-15.0) | - | 4.5
(0.8-15.0) | - |
| GPT-4.1 | Manual Impl (Isolated Code Interp) | 0.72±0.03 | 50.3
(10.9-364.9) | 0.0
(0.0-0.0) | 50.3
(10.9-365.0) | 2.8
(1.0-21.0) |
| GPT-4.1 | NIGHTJAR (Baseline) | 0.61±0.02 | 39.0
(4.7-957.4) | 0.0
(0.0-0.0) | 39.0
(4.7-957.4) | 24.9
(4-162) |
| GPT-4.1 | NIGHTJAR (Isolated Eval) | 0.61±0.02 | 51.7
(7.8-282.1) | 0.0
(0.0-0.0) | 51.7
(7.8-282.1) | 32.3
(6.0-174.0) |
| GPT-4.1 | NIGHTJAR (Shared Eval) | 0.62±0.08 | 75.0
(6.9-579.4) | 0.0
(0.0-0.0) | 75.0
(6.9-579.4) | 49.2
(6.0-289.0) |
| GPT-4.1 | NIGHTJAR (Shared Python) | 0.81±0.04 | 71.8
(4.0-417.0) | 0.0
(0.0-0.0) | 71.8
(4.0-417.0) | 63.1
(2-299) |
| GPT-4.1 | NIGHTJAR (Shared Python, Caching) | 0.78±0.03 | 66.2
(7.9-406.1) | 0.0
(0.0-0.0) | 66.2
(7.9-406.1) | 68.6
(7-300) |
| GPT-4.1 | NIGHTJAR | 0.78±0.03 | 19.6
(3.0-72.1) | 0.0
(0.0-0.1) | 19.7
(3.1-72.1) | 16.6
(2-96) |
| GPT-4.1 | LLM Code Generation | 0.45±0.02 | 2.8
(0.0-31.8) | 5.0
(0.9-23.7) | 7.7
(0.9-36.9) | - |

Table 5: Performance results of subset of *SPSBench* that does not require first-order functions or abstract methods. See Appendix C for which programs are in this subset. Average pass rate and runtime with ranges in gray.

| Model | Method | Pass Rate | Runtime (s) | Compile Time (s) | Total Time (s) | # Effects |
|---|---|---|---|---|---|---|
| Sonnet 4 | Manual Impl | 0.78±0.04 | 8.2 (1.3-44.8) | - | 8.2 (1.3-44.8) | - |
| Sonnet 4 | Manual Impl (Isolated Code Interp) | 0.63±0.04 | 42.5 (3.2-171.5) | 0.0 (0.0-0.0) | 42.5 (3.2-171.5) | 3.1 (0-14) |
| Sonnet 4 | NIGHTJAR (Baseline) | 0.69±0.05 | 57.8 (9.6-721.0) | 0.0 (0.0-0.0) | 57.8 (9.6-721.0) | 29.6 (6-300) |
| Sonnet 4 | NIGHTJAR (Isolated Eval) | 0.76±0.04 | 103.2 (14.9-870.0) | 0.0 (0.0-0.0) | 103.2 (14.9-870.0) | 36.2 (6-300) |
| Sonnet 4 | NIGHTJAR (Shared Eval) | 0.85±0.03 | 104.1 (10.6-464.2) | 0.0 (0.0-0.0) | 104.2 (10.6-464.2) | 36.6 (6-119) |
| Sonnet 4 | NIGHTJAR (Shared Python) | 0.83±0.04 | 85.6 (21.6-236.4) | 0.0 (0.0-0.0) | 85.6 (21.6-236.5) | 29.2 (7-93) |
| Sonnet 4 | NIGHTJAR (Shared Python, Caching) | 0.83±0.03 | 75.6 (19.0-244.8) | 0.0 (0.0-0.0) | 75.7 (19.0-244.8) | 30.5 (8-106) |
| Sonnet 4 | NIGHTJAR | 0.88±0.03 | 29.3 (9.1-83.0) | 0.0 (0.0-0.0) | 29.3 (9.1-83.1) | 10.3 (4-33) |
| Sonnet 4 | LLM Code Generation | 0.41±0.05 | 7.9 (0.0-38.6) | 7.5 (3.6-17.8) | 15.4 (3.6-55.5) | - |
| GPT-4.1 | Manual Impl | 0.75±0.04 | 4.6 (0.8-15.0) | - | 4.6 (0.8-15.0) | - |
| GPT-4.1 | Manual Impl (Isolated Code Interp) | 0.71±0.03 | 52.5 (10.9-364.9) | 0.0 (0.0-0.0) | 52.5 (10.9-365.0) | 3.0 (1.0-21.0) |
| GPT-4.1 | NIGHTJAR (Baseline) | 0.70±0.03 | 44.9 (4.7-957.4) | 0.0 (0.0-0.0) | 44.9 (4.7-957.4) | 27.7 (4-162) |
| GPT-4.1 | NIGHTJAR (Isolated Eval) | 0.72±0.04 | 55.0 (7.8-282.1) | 0.0 (0.0-0.0) | 55.0 (7.8-282.1) | 35.3 (6.0-174.0) |
| GPT-4.1 | NIGHTJAR (Shared Eval) | 0.70±0.04 | 71.7 (6.9-579.4) | 0.0 (0.0-0.0) | 71.7 (6.9-579.4) | 48.6 (6.0-289.0) |
| GPT-4.1 | NIGHTJAR (Shared Python) | 0.79±0.05 | 79.9 (11.4-417.0) | 0.0 (0.0-0.0) | 79.9 (11.5-417.0) | 67.2 (12-299) |
| GPT-4.1 | NIGHTJAR (Shared Python, Caching) | 0.76±0.05 | 68.4 (11.2-406.1) | 0.0 (0.0-0.0) | 68.4 (11.2-406.1) | 70.9 (11-300) |
| GPT-4.1 | NIGHTJAR | 0.76±0.02 | 22.7 (3.3-72.1) | 0.0 (0.0-0.1) | 22.7 (3.3-72.1) | 18.3 (2-96) |
| GPT-4.1 | LLM Code Generation | 0.48±0.06 | 0.3 (0.0-1.8) | 5.0 (1.6-13.4) | 5.3 (1.6-13.4) | - |

## E.3 LOCALLY HOSTED LLM

We evaluate the performance of NIGHTJAR using a smaller, locally hosted model as the base LLM on *SPSBench*. We used GPT-OSS 20B served through VLLM (Kwon et al., 2023) on an A100 80GB GPU. Table 6 shows the average pass rate with standard deviation, the average runtime with ranges, and the average number of emitted effects with ranges over 5 runs. Using GPT-OSS 20B, the pass rate of NIGHTJAR is lower than the manual implementation. This is because the NIGHTJAR implementation relies heavily on the agentic capabilities of the base LLM. Further system prompt engineering, optimization, or finetuning could improve the pass rate when using a smaller model. Nevertheless, NIGHTJAR is still more performant than using an LLM to statically generate Python code (*LLM Compiler*).

Table 6: Average pass rate, runtime, compile time, and number of effects using GPT-OSS 20B with ranges shown in gray.

| Method | Pass Rate | Runtime (s) | Compile Time (s) | Total Time | # Effects |
|---|---|---|---|---|---|
| Manual Impl | 0.68±0.04 | 5.3 (0.3-89.2) | - | 5.3 (0.3-89.2) | - |
| LLM Compiler | 0.13±0.03 | 1.2 (0.00-55.24) | 4.3 (0.39-16.46) | 5.3 (0.39-57.78) | - |
| NIGHTJAR (Shared Python) | 0.47±0.03 | 54.2 (3.2-581.8) | 0.0 (0.0-0.0) | 54.2 (3.2-581.8) | 28.1 (2-242) |
| NIGHTJAR | 0.40±0.02 | 36.5 (1.4-385.7) | 0.0 (0.0-0.0) | 36.5 (1.45-385.68) | 8.0 (0-46) |

### E.4 TEMPERATURE 0

NIGHTJAR uses closed LLMs and temperature 1 for performant LLM agents, which means program executions may be non-deterministic. We performed an ablation of the evaluation with temperature of 0 on GPT-4.1. Table 7 shows the average pass rate with standard deviation and average runtime with ranges over 5 runs. As shown, NIGHTJAR also achieves parity in pass rate to manual implementations with temperature 0. We note that even using temperature 0 does not result in deterministic executions because the base LLM is a closed model (He & Lab, 2025). Using temperature 0 on a locally-hosted open model and controlling for LLM non-determinism (He & Lab, 2025) or caching and re-using responses to the same effect trace would enable deterministic NIGHTJAR executions.

Table 7: Average pass rate and runtime with ranges in gray.

| Model | Method | Pass Rate | Time (s) | # Effects |
|---|---|---|---|---|
| GPT-4.1 | Manual Impl | 0.74±0.01 | 5.0 (0.8-23.0) | - |
| GPT-4.1 | NIGHTJAR (Shared Python) | 0.73±0.03 | 82.9 (3.4-548.6) | 79.9 (4.0-300.0) |
| GPT-4.1 | NIGHTJAR | 0.77±0.04 | 28.9 (2.9-267.3) | 20.2 (2.0-149.0) |

### E.5 LARGE DATA STRUCTURE

The representation of program data as they are accessed by an LLM factors into the program performance enabled by a programming system. There are two methods to pass program values to and from LLMs: *pass-by-copy* and *pass-by-reference*.

Program values are typically encoded as strings in JSON format and included in the prompt context for the LLM to process. This method is called pass-by-copy. It is intractable when computing with large objects, because LLMs have a limited context length. The length of the encoded object string is linear to the size of the object, so a sufficiently large object would be beyond the context length of the LLM. This is also the case for updating or writing an object, where the size of objects the LLM can compute with is limited by the maximum output length.

In contrast, pass-by-reference avoids the need to encode objects into strings. For an LLM to read objects, it first receives a reference to the object, which is only a few digits in length and constant in the size of the object. During program execution, the LLM inspects only the relevant parts of the object it needs for the prompt evaluation. This is more efficient when the prompt requires the use of only a part of the large object.

We examine how pass-by-reference affects performance compared to pass-by-copy with a case study using the Figure 1 programs. Figure 1a is implemented manually by a programmer. It uses pass-by-copy by encoding the graph object into JSON format (Line 10) and passing it to the LLM in the prompt (Line 34). The output graph is also decoded via JSON format and reified into the `Graph` object (Lines 11 to 15).

**Methodology.** We compare the performance of executing the Figure 1b with 6 different natural language queries (Appendix A.2) using NIGHTJAR and NIGHTJAR (Baseline) against the performance of executing the pass-by-copy implementation by a programmer (the full Figure 1a program; Appendix A.1). With shared heap, NIGHTJAR (Baseline) uses pass-by-reference to represent mutable data structures NIGHTJAR also uses pass-by-reference, but also additionally enables the use of Python functions. We execute each program on generated graphs of exponentially increasing sizes 5 times each, using GPT-4.1-2025-04-14 on a machine with an M1 Max chip and 64GB memory. Due to cost constraints, we set a limit of 100 steps per query for the NIGHTJAR agent loop.

**Results.** Figure 10 shows the performance. The pass-by-copy implementation generally achieves lower accuracy than NIGHTJAR implementations. On small graphs with fewer than 100 nodes, the pass-by-copy implementation generally achieves lower execution time, lower token usage, and lower cost. Since the graph is encoded into JSON format and appended into the prompt, the number of input tokens increases linearly with the number of nodes. After 1000 nodes, the encoded graph exceeds the context window size, resulting in failure to answer any queries. The execution time, output token usage, and cost also increase as the graph size increases. In comparison, NIGHTJAR

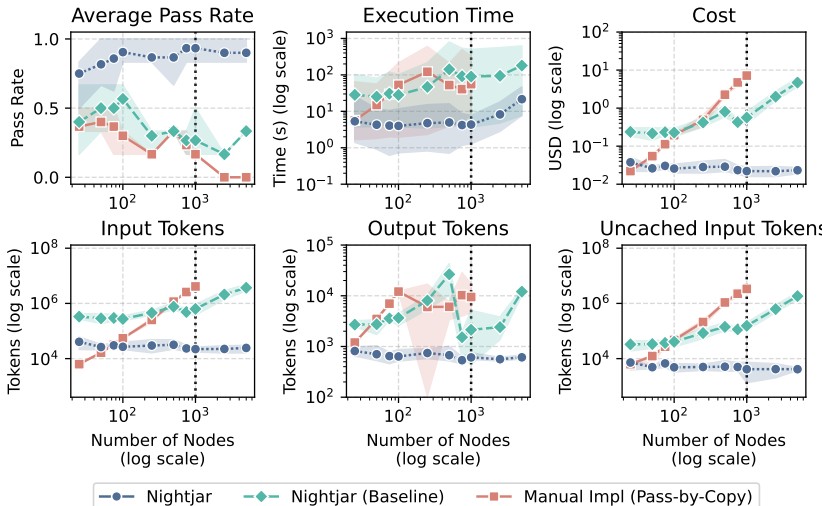

Figure 10: Performance of the Figure 1 programs. The shaded area shows the range of values. The black dotted line marks when the encoded graph exceeds the context window size with pass-by-copy.

and NIGHTJAR (Baseline), using pass-by-reference to represent mutable objects, query pieces of the graph as necessary. Thus, the token usage and execution time of NIGHTJAR and NIGHTJAR (Baseline) increase at a slower rate compared to the pass-by-copy implementation as the graph size increases. This enables NIGHTJAR (Baseline) to answer queries on large graphs when the pass-by-copy implementation cannot, resulting in the higher pass rates. The specialization to Python of NIGHTJAR enables the use of Python functions, which improves further the pass rate over the baseline implementation. While programmers can manually implement using pass-by-reference, NIGHTJAR supports this data representation style out of the box without additional configuration by the programmer.

### E.6 NUMBER OF EFFECTS TO RUNTIME

There is a direct relation between program runtime executed using NIGHTJAR with the number of effects used, as each emitted effect corresponds to one inference pass through the base LLM. Figure 11 is a scatterplot of the number of effects used and the runtime of each program execution of *SPSBench*. It shows that the program runtime is proportional to the number of effects emitted during program execution, with different LLMs exhibiting different ratios of runtime to number of emitted effects.

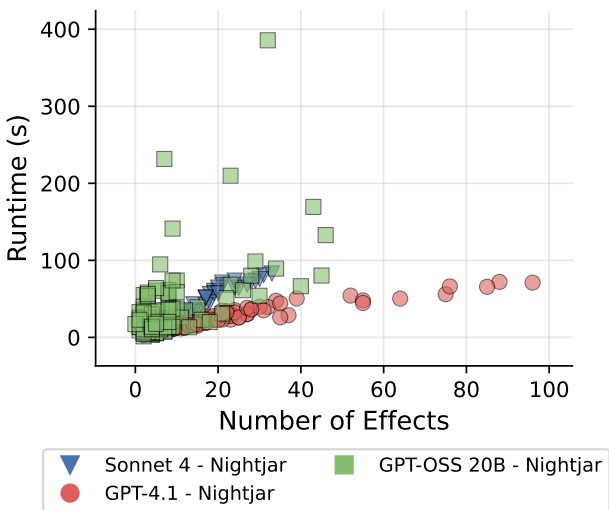

Figure 11: Number of emitted effects in a program execution to runtime.

### E.7 ADDITIONAL RESULTS ON LLM MATH REASONING BENCHMARK

We evaluate the performance of NIGHTJAR on the full evaluation set of GMS8K (Cobbe et al., 2021), consisting of 1319 math word problems using GPT-4.1. GSM8K is a benchmark for evaluating LLM math reasoning, and thus does not require complex data structures nor meaningfully use formal program state to solve. In this sort of application, using NIGHTJAR is equivalent to using an LLM agent with code evaluation tools with isolated program states. We evaluate the performance of NIGHTJAR on GSM8K to empirically validate this and show that NIGHTJAR's performance is competitive.

We compare against the manual implementation of program, which queries the LLM for a string expression that the program then invokes the Python `eval` function on. We also compare against a version of manual implementation that employs tool use with the OpenAI Code Interpreter (which evaluates code in an isolated container) and a version that implements tool use with a custom, user-defined code evaluation tool that maintains isolated program states.

Table 8 shows the pass rate, the average runtime, and the average number of tool calls or effects over 3 runs. NIGHTJAR achieves higher or equal accuracy to the manual implementations, demonstrating that the NIGHTJAR system and the shared program state setup do not detract from LLM capabilities on standard LLM tasks. NIGHTJAR incurs slightly higher runtime overhead than the basic manual implementation program, because NIGHTJAR requires the LLM to emit effects. However, this runtime overhead is similar to the overhead incurred by utilizing tool use with the OpenAI code interpreter tool. Additionally, NIGHTJAR's runtime overhead can be reduced with further system engineering as shown by the optimized version of NIGHTJAR.

Table 8: GSM8K results using GPT-4.1.

| Method | Pass Rate | Avg Time (s) | # Tool Calls or Effects |
|---|---|---|---|
| Manual Impl | 0.84±0.01 | **1.4** (0.8-50.4) | - |
| Manual Impl w/ OpenAI Code Interpreter | **0.92±0.01** | 5.1 (0.1-304.7) | 0.9 (0-3) |
| Manual Impl w/ Custom Code Interpreter | 0.70±0.01 | 2.6 (1.0-24.9) | 1.6 (0-4) |
| NIGHTJAR (Shared Python) | **0.93±0.00** | 5.9 (2.3-58.2) | 6.6 (0-29) |
| NIGHTJAR | **0.93±0.00** | 4.7 (1.7-36.3) | 4.1 (0-16) |

## F  FAILURE ANALYSIS

We analyzed the 125 effect traces of NIGHTJAR on execution programs from *SPSBench* using GPT-4.1 and Sonnet 4. We labeled the reason the LLM failed to get a perfect score on the program for each run of the program (25 program x 5 runs each = 125 runs in total). Table 9 shows the summary. Most program runs pass all test cases (*100% Pass*). It is only in a few instances that the LLM gives up on evaluating a prompt when it encounters an error (*Give Up*), hallucinates data (*Hallucination*), or incorrectly performed the program state operation specified in the prompt (*Incorrect State Op*). Most of the time, the LLM fails test cases due to making incorrect reasoning conclusions or math computations (*Reasoning*). Decreasing failures due to incorrect reasoning requires more capable LLM models or additional prompt engineering on the program's prompts or the LLM system prompt.

Table 9: NIGHTJAR failure analysis summary.

| Category | Sonnet 4 | GPT-4.1 |
|---|---|---|
| 100% Pass | 82 (65.6%) | 69 (55.2%) |
| Reasoning | 41 (32.8%) | 47 (37.6%) |
| Give Up | 0 (0%) | 4 (3.2%) |
| Incorrect State Op | 1 (0.8%) | 5 (4.0%) |
| Hallucination | 1 (0.8%) | 0 (0%) |
| Total | 125 | 125 |

We also further analyzed the types of error messages the LLM encounters when issuing effects and how it attempts to recover from these errors. Table 10 and Table 11 show the summaries. Sonnet 4 encountered very few errors, most of which are due to incorrect syntax or incorrect data type assumptions. GPT-4.1 encountered many more errors, most of them are due to confusing which effects to use (*State Op Error*). For example, it often tried to use EVAL instead of EXEC, before correcting its mistake to use EXEC successfully. It also often hallucinates variables (*Forbidden Var*) by confusing references for variable names or using _ as a placeholder inappropriately. When encountering errors, the LLM generally corrects its mistake. GPT-4.1 sometimes ignores the error and performs a different operation instead or finds a workaround to performing the same computation.

Table 10: Types of errors encountered by emitted effects.

| Error Category | Sonnet 4 | GPT-4.1 |
|---|---|---|
| State Op Error | 0 (0%) | 219 (67.8%) |
| Forbidden Var | 0 (0.0%) | 70 (21.7%) |
| Type Error | 4 (36.4%) | 15 (4.6%) |
| Syntax Error | 4 (36.4%) | 13 (4.0%) |
| Function Call Error | 2 (18.2%) | 2 (0.6%) |
| Index Error | 1 (9.1%) | 4 (1.2%) |
| Total | 11 | 323 |

Table 11: Method of recovery by the LLM when encountering an error during effect handling.

| Recovery Method | Sonnet 4 | GPT-4.1 |
|---|---|---|
| Correction | 7 (63.6%) | 199 (61.6%) |
| Ignored Error | 1 (9.1%) | 59 (18.3%) |
| Workaround | 2 (18.2%) | 34 (10.5%) |
| Retry | 0 (0%) | 22 (6.8%) |
| Not Recovered | 1 (9.1%) | 9 (2.8%) |
| Total | 11 | 323 |

We show below the effect trace of the example program (*graph* in *SPSBench*) that did not pass the test case with the input query "update the graph so paper 5 cites paper 14." The execution did not

correctly update the graph as it confused the direction of the graph edges, and instead updated the graph such that paper 14 cites paper 5:

```
Effect: eval(str(type(query)))
Result: "<class 'str'>"
Effect: eval(str(type(graph)))
Result: "<class '<run_path>.Graph'>"
Effect: eval(hasattr(graph, '__doc__'))
Result: True
Effect: eval(getattr(graph, '__doc__'))
Result: "A directed graph. Nodes are represented by a set of node values. Edges are represented by a
    dictionary of source node value to a set of target node values."
Effect: eval(str(dir(graph)))
Result: "['__annotations__', '__class__', '__contains__', '__delattr__', '__dict__', '__dir__', '__doc__
    ', '__eq__', '__firstlineno__', '__format__', '__ge__', '__getattribute__', '__getstate__', '__gt__
    ', '__hash__', '__init__', '__init_subclass__', '__le__', '__len__', '__lt__', '__module__', '
    __ne__', '__new__', '__reduce__', '__reduce_ex__', '__repr__', '__setattr__', '__sizeof__', '
    __slotnames__', '__static_attributes__', '__str__', '__subclasshook__', '__weakref__', 'edges', '
    nodes']"
Effect: eval(str(type(query.strip().lower())))
Result: "<class 'str'>"
Effect: eval(query.strip().lower())
Result: "update the graph so paper 5 cites paper 14."
Effect: eval(graph.edges)
Result: Ref(1018[Obj])
Effect: eval(str(graph.edges))
Result: "{16: {4, 5, 10, 11, 17, 19, 21}, 12: {1, 5, 6, 7, 9, 10, 11, 14, 15, 17, 18, 19, 23, 24}, 9: {1,
    2, 3, 4, 5, 6, 7, 10, 15, 20, 21, 22, 23}, 19: {0, 1, 4, 5, 10, 14, 15, 17, 20}, 18: {0, 3, 5, 6,
    17, 20, 21, 22, 23}, 6: {3, 4, 8, 11, 13, 14, 20, 21, 22, 24}, 5: {2, 3, 7, 8, 11, 13, 14, 21, 23,
    24}, 10: {0, 1, 2, 3, 7, 8, 11, 14, 21, 24}, 15: {0, 1, 2, 4, 7, 13, 14, 17, 23, 24}, 21: {2, 3, 4,
    7, 8, 11, 23, 24}, 11: {1, 3, 4, 17, 23}, 17: {2, 3, 7, 8, 22, 23}, 1: {3, 4, 7, 8, 20, 22}, 14:
    {2, 7, 8, 13, 24}, 22: {0, 2, 20, 13}, 13: {24, 0, 4, 7}, 2: {3, 7, 20, 23, 24}, 23: {24, 0, 4}, 4:
    {24, 8, 7}, 24: {0, 3, 20}, 8: {0, 3, 20}, 3: {20}}"
Effect: eval(str(graph.nodes))
Result: "{0, 1, 2, 3, 4, 5, 6, 7, 8, 9, 10, 11, 12, 13, 14, 15, 16, 17, 18, 19, 20, 21, 22, 23, 24}"
Effect: eval(14 in graph.nodes and 5 in graph.nodes)
Result: True
Effect: eval(5 in graph.edges)
Result: True
Effect: eval(14 in graph.edges[5])
Result: True
Effect: eval(set(graph.edges[5]))
Result: Ref(1116[Obj])
Effect: eval(sorted(list(graph.edges[5])))
Result: Ref(1152[Obj])
Effect: eval(str(sorted(list(graph.edges[5]))))
Result: "[2, 3, 7, 8, 11, 13, 14, 21, 23, 24]"
Effect: eval('break' in query.strip().lower() or 'terminate' in query.strip().lower())
Result: False
Effect: eval(sorted(list(graph.edges[5])))
Result: Ref(1372[Obj])
Effect: eval(str(sorted(list(graph.edges[5]))))
Result: [2, 3, 7, 8, 11, 13, 14, 21, 23, 24]
Effect: eval(response = sorted(list(graph.edges[5])))
Result: Success
Effect: done()
```

## G  LLM USAGE DISCLOSURE

We used LLMs to correct grammar, to find related works, and to brainstorm conceptual terminology.

