# OpenReview forum: "Sharing State Between Prompts and Programs"
_ICLR.cc/2026/Conference — ICLR 2026 Poster_

### Official Review · Reviewer_KmWC · 2025-10-31

**Soundness:** 3
**Presentation:** 3
**Contribution:** 2
**Rating:** 4
**Confidence:** 3

**Summary:**

The paper introduces **Shared Program State (SPS)**, a novel abstraction allowing natural-language code (prompts) and traditional programming languages (like Python) to **share variables, objects and control flow**. It defines a schema called the Natural Function Interface (NFI) that models this sharing via operations like `Lookup`, `Assign`, `Goto` etc. The authors implement this in a system called NIGHTJAR, which co-executes Python and natural code. Empirical evaluation (on a 25-program benchmark) shows comparable accuracy with much shorter code (23-82% fewer lines), albeit with higher runtime overhead.

**Key Contributions**
1. Definition of the Shared Program State abstraction and its semantics for mixing natural and formal code.
2. Formalization of the Natural Function Interface (NFI) schema for the interaction.
3. Implementation (NIGHTJAR) demonstrating how natural language code can manipulate program state in a host language.
4. Empirical study (SPSBench) showing productivity gains (code conciseness) and feasibility of SPS, with noted runtime cost.

**Strengths:**

* The paper introduces a novel abstraction; the idea that prompts (natural-language code) and traditional programming languages can share program state; variables, heap objects, control flow. This moves beyond prior work that treated prompts as black-box tool calls or code generation endpoints. For example, [A] considers prompts as programs but does not formalize shared variable/state semantics between NL and formal code). [B] supports collaborative prompt engineering, but again focuses on prompt sharing/management rather than state-sharing semantics between NL and host code.
* The combination of techniques is also creative: using an LLM as a "natural interpreter", defining an interface (Natural Function Interface) that models state-manipulation effects (assign, lookup, goto) and handlers in a host language. Prior works on NL to code generation (e.g., prompt-based learning, chain-of-thought) separate the code generation from program execution; here the "code" (NL) directly manipulates execution state. For instance, the "Program of Thoughts" [C] work uses LMs to generate a program which is then executed externally (i.e., separate stages). In contrast, this paper merges the NL/code boundary via shared state. That creative combination marks a strong originality.
* The clarity of the interface model (Natural Function Interface), and the way the paper formalises values/effects/handlers, suggests technical rigor. This goes beyond more HCI-oriented prompt papers (e.g., CoPrompt [B]) which focus on usability rather than formal semantics.

```
[A] Prompts Are Programs Too! Understanding How Developers Build Software Containing Prompts, FSE 2025
[B] CoPrompt: Supporting Prompt Sharing and Referring in Collaborative Natural Language Programming, CHI 2024
[C] Program of Thoughts Prompting: Disentangling Computation from Reasoning for Numerical Reasoning Tasks, TMLR 2023
```

**Weaknesses:**

* The empirical evaluation uses a 25-program suite (e.g., SPSBench) that demonstrates the abstraction in a limited number of tasks. It is unclear how well the approach scales to larger, more complex programs, especially ones with deeply nested state, concurrency, or rich object graphs.
* The implementation incurs a substantial runtime penalty (11.7–15.3× slower) due to the involvement of an LLM interpreting natural code. While the paper acknowledges it, the evaluation does not fully explore how this overhead affects usability in real-world settings. Many practitioners may find this cost prohibitive.
* Allowing natural-language code to manipulate host program state (variables, heap, control flow) introduces risks: unintended side-effects, state inconsistency, security holes, debugging difficulty. The paper acknowledges safety issues but does not deeply evaluate them (e.g., what happens if the NL code assigns to a variable incorrectly, or dereferences invalid references).
* While the “shared program state” abstraction is compelling, some of its components (NL interfacing with program state, prompts-as-programs) overlap substantially with prior work. For instance, [A] already frames prompts as programs. The novelty claim would be stronger if the paper explicitly differentiates itself from these works and shows how state-sharing goes beyond "prompts as programs" and prior tool-calling frameworks.
* Although the formalism is strong, there may be a steep learning curve for typical developers: understanding effects/handlers, bridging NL syntax with host language semantics, maintaining coherence in hybrid code. The paper could provide more empirical evidence on developer usability: how quickly developers adapt to writing natural code that manipulates host state, what errors or misunderstandings occur. In contrast, works like CoPrompt [B] focus on developer workflows and collaboration in prompt engineering and report usability metrics.
```
[A] Prompts Are Programs Too! Understanding How Developers Build Software Containing Prompts, FSE 2025
[B] CoPrompt: Supporting Prompt Sharing and Referring in Collaborative Natural Language Programming, CHI 2024
```

**Questions:**

* Is it possible to include a larger scale benchmark with more complex stateful programs (e.g., classes, inheritance, mutable collections, parallel threads) and evaluate how the shared state abstraction performs there (both correctness and latency)?
* Is it possible to compare the performance (accuracy, conciseness, runtime) on tasks used in ANPL [A] or similar to increase external validity?
* Is it possible to provide a deeper latency cost-benefit analysis (e.g., user-perceived delay, scalability with number of state operations, caching strategies for repeated NL manipulations)?
* Is it possible to include a dedicated "robustness" evaluation: fault injection (NL commands that try invalid operations), tracking state invariants, measuring how many errors occur, how they are detected and recovered? Also are there guidelines or restrictions on NL code allowed (e.g., sandboxing, type checks, access control)?
* Is it possible to add a discussion on how debugging mixed NL+code state will be supported (logging, visualization, stepping)?
* Is it possible to deepen the literature comparison section: explicitly list prior abstractions (e.g., prompt-programming, tool-calling interfaces, NL-to-code) and highlight exactly how the shared -state model differs (e.g., direct variable reference, control flow manipulation)?
* Is it possible to include a user study (even small-scale) with software engineers or prompt-programmers: measure how well they'd adopt the shared-state model, what errors are common, what cognitive load is involved?

```
[A] ANPL: Towards Natural Programming with Interactive Decomposition, NeurIPS 2023
```

---

> ### Author Response · Authors · 2025-11-21
>
> We thank the reviewer for the thoughtful comments and questions. We address each of them below:
>
> > It is unclear how well the approach scales to larger, more complex programs, especially ones with deeply nested state, concurrency, or rich object graphs.
>
> > Is it possible to include a larger scale benchmark with more complex stateful programs (e.g., classes, inheritance, mutable collections, parallel threads) and evaluate how the shared state abstraction performs there (both correctness and latency)?
>
> Several of our benchmarks involve classes, inheritance, and mutable collections. For example, the `stores` program require defining subclasses and creating instances of those subclasses.
> ```
> from abc import ABC, abstractmethod
> from typing import Tuple, List, Any, Dict
>
> class Store(ABC):
>     def __init__(self, name: str, city: str):
>         self.name = name
>         self.city = city
>
>     def get_description(self):
>         return f"{self.name}, located in {self.city}"
>
>     @abstractmethod
>     def get_inventory(self) -> List[str]: ...
>
> def main(prompt: str):
>     """natural
>     Read the <prompt> and extract what kinds of <Store>s are described.
>
>     For each identified store type:
>     - Create a subclass of <Store> with a custom `get_inventory` method
>     - Instantiate a store of the subclass with appropriate name and city based on the prompt
>     - Store the store instances in a list called <:stores>
>     """
>
>     return stores
> ```
> `battle`, `calendar`, `enemy`, `fact_checking`, `graph`, `menu`, `task_completion`, `tiny_bookshop` all involve mutable collections.
>
> > While the paper acknowledges it, the evaluation does not fully explore how this overhead affects usability in real-world settings. Many practitioners may find this cost prohibitive.
>
> We expect that future work on optimizing and engineering programming systems enabling shared program state will reduce the runtime costs to the abstraction. We demonstrate the viability of this line of future work by implementing a version of Nightjar in which the system eagerly gives the type, heap references, and values of `<var>`-denoted variables to the LLM along with the natural code to execute. We refer to this version as `Optimized`. We show below the average pass rate with standard deviation, the average runtime, and the average number of effects used across 5 runs. The optimization maintains the pass rate and reduces the runtime by 2.9-4.0x from the vanilla Nightjar implementation. Specifically, the optimization reduces the overhead of waiting for the LLM to emit effects to read the program state, since it is almost always the case that the LLM should read `<var>`-denoted variables in the natural code. This is shown by the reduced average number of effects used by the Optimized Nightjar. We have included these results in the Evaluation section of the revised draft.
>
> | Model    | Method                      | Pass Rate | Runtime |
> |----------|-----------------------------|-----------|---------|
> | Sonnet 4 | Programmer Impl             | 0.78±0.03 | 7.97    |
> | Sonnet 4 | Nightjar (Ours)             | 0.85±0.02 | 75.80   |
> | Sonnet 4 | Nightjar (Ours) (Optimized) | 0.85±0.03 | 25.90   |
> | GPT-4.1  | Programmer Impl             | 0.74±0.03 | 4.53    |
> | GPT-4.1  | Nightjar (Ours)             | 0.81±0.04 | 71.83   |
> | GPT-4.1  | Nightjar (Ours) (Optimized) | 0.78±0.03 | 19.64   |
>
> *Note: We corrected bugs in `balanced_parenthesis`, `calendar`, `coffeeshop`, `email_classification`, and `stores` that improved pass rates for all techniques versus the original paper, but still preserved the conclusions. We have updated the result table in the revised draft accordingly.
>
> > [A] already frames prompts as programs. The novelty claim would be stronger if the paper explicitly differentiates itself from these works and shows how state-sharing goes beyond "prompts as programs" and prior tool-calling frameworks.
>
> We would like to clarify that we do not claim framing "prompts as programs" as a contribution. Our contribution is in presenting the shared program state programming model and developing a formal system specification and implementation for enabling shared program state. No prior work enables the shared program state programming model.
>
> > Is it possible to deepen the literature comparison section: explicitly list prior abstractions (e.g., prompt-programming, tool-calling interfaces, NL-to-code) and highlight exactly how the shared -state model differs (e.g., direct variable reference, control flow manipulation)?
>
> In the Related Work section of the revised draft, we have explicitly listed prior abstractions as paragraph headings and incorporated the mentioned prior works. We added further elaborations from the original draft on the differences between shared program state and prior work to the revised draft as well.

---

> ### Author Response · Authors · 2025-11-21
>
> > there may be a steep learning curve for typical developers: understanding effects/handlers, bridging NL syntax with host language semantics, maintaining coherence in hybrid code.
>
> Effects are not visible to program developers. Section 4.2 of the paper illustrates the full extent of the syntax as a concept a program developer needs to understand to write programs. Effects/handlers are only relevant to those who contribute to Nightjar or who desire to port Nightjar to another language beyond Python. The typical program developer only needs to write programs similar to Figure 1a) and execute it.
>
> The only syntax needed with Nightjar is using `"""natural[...]"""` to demark natural code, use `<var>` to refer to Python variables, and to use `<:var>` to denote writing to Python variables. The need of program developers to maintain coherence in hybrid code is the same as they would need to write the program by hand without shared program state.
>
> > The paper could provide more empirical evidence on developer usability
>
> > Is it possible to include a user study
>
> Our evaluation strategy is consistent with prior published works in ICLR and NeuRIPS related to enabling interoperability between natural code and formal programs such as DSPy[1] and SGLang[2]. Moreover, in addition to publication, both of these systems have seen significant public adoption since publication and neither performs a user study. These papers instead focus on the technical underpinnings of the programming model, as we do in our paper. As such, a complete user study on the many different frameworks for embedding natural code in formal programs is beyond the scope of this work.
>
> > Is it possible to compare the performance (accuracy, conciseness, runtime) on tasks used in ANPL [A] or similar to increase external validity?
>
> While programmers are increasingly integrating prompts into programs, no existing benchmarks that we know of tests the interoperability of programs with embedded natural code. Existing LLM benchmarks such as GSM8k[3], GPQA[4], HotPotQA[5], ARC[6] test the inherent reasoning and problem solving capabilities of LLMs; they are not designed to test the performance of a program that utilizes both formal program constructs/data structures and natural code.
>
> As these existing benchmarks do not require the use of a formal program state at all, they do not test the capability of an LLM system to perform operations on the program state. This is why we developed the SPSBench benchmark suite.
>
> For example, a test problem from GSM8K is
> ```
> A robe takes 2 bolts of blue fiber and half that much white fiber. How many bolts in total does it take?
> ```
> Notably, this test problem is entirely natural code and does not meaningfully utilize a formal program state, because GSM8K is fundamentally a benchmark for evaluating an LLM's math reasoning capability, not natural-formal code interoperability.
>
> We have further evaluated on all 1319 problems from the GSM8K eval set using GPT-4.1 as the LLM. We compare against the programmer implementation program, which queries the LLM for a string expression that it invokes the Python `eval` function on. We also compare against a version of programmer implementation that uses tool use with the OpenAI Code Interpreter (which evaluates code in an isolated container) and a version that implements tool use with a custom, user-defined code evaluation tool that maintain isolated program states.
>
>
> We show the average pass rate, runtime, and number of tool calls or effects over 3 runs. As shown, Nightjar achieves higher or equal accuracy to the programmer implementations, demonstrating that the setup does not detract from LLM reasoning capabilities. Nightjar incurs slight runtime overhead due to emitting effects, but 1) the runtime is approximately the same as using OpenAI's code interpreter tool in the programmer implementation and 2) the overhead can be reduced with engineering efforts as shown by Optimized Nightjar. We have added these results to `Appendix L` of the revised draft.
>
> | Method                      | Pass Rate | Runtime | # Tool Calls/Effects |
> |-----------------------------|-----------|---------|----------------------|
> | GPT-4.1  | Programmer Impl                    | 0.84±0.01   | 1.4    | 0.0     |
> | GPT-4.1  | Programmer Impl  w/ Code Interpreter  | 0.92±0.01   | 5.1   | 0.9     |
> | GPT-4.1  | Programmer Impl w/  Custom Code Interpreter | 0.70±0.01   | 2.6    | 1.6     |
> | GPT-4.1  | Nightjar (Ours)                    | 0.93±0.00   | 5.9    | 6.6    |
> | GPT-4.1  | Nightjar (Ours) (Optimized)        | 0.93±0.00   | 4.7    | 4.1    |

---

> > ### Author Response · Authors · 2025-11-21
> >
> > > caching strategies for repeated NL manipulations
> >
> > For GPT-4.1, OpenAI automatically enables prompt caching, and we also make use of Anthropic's prompt caching feature for Sonnet 4. We additionally implemented a simple scheme of caching effects when given the same effect trace. This means that 1) until the LLM inspects program inputs, the same effects will be emitted, and 2) given the same natural code and the same program inputs, the same effect trace will be used. To maintain stochasticity between runs, we clear the cache per program run, but still allowing different test cases to share cache. We show below the average pass rate with standard deviation, average runtime, and average number of effects across 5 runs. This resulted in some reduction in the runtime. We leave more complicated caching strategies and other system design explorations for future work.
> >
> > | Model    | Method                      | Pass Rate | Runtime | Effects    |
> > |----------|-----------------------------|-----------|---------|------------|
> > | Sonnet 4 | Nightjar (Ours)             | 0.85±0.02 | 75.80   | 27.32      |
> > | Sonnet 4 | Nightjar (Ours) (Caching)   | 0.84±0.02 | 70.63   | 27.91      |
> > | GPT-4.1  | Nightjar (Ours)             | 0.81±0.04 | 71.83   | 63.14      |
> > | GPT-4.1  | Nightjar (Ours) (Caching)   | 0.78±0.03 | 66.23   | 68.64      |
> >
> > *Note: We corrected bugs in `balanced_parenthesis`, `calendar`, `coffeeshop`, `email_classification`, and `stores` that improved pass rates for all techniques versus the original paper, but still preserved the conclusions. We have updated the result table in the revised draft accordingly.
> >
> > > scalability with number of state operations,
> >
> > We include in `Appendix K` of the revised draft a scatter plot of the number of effects to the runtime of each program run. The more number of effects the program uses, the longer the runtime. However, the ratio of latency to number of effects varies by the model.
> >
> > > user-perceived delay
> >
> > We are unclear on what the reviewer means by user-perceived delay. Could the reviewer clarify what the analysis requested consists of?
> >
> > > type checks, access control
> >
> > As described in Section 5, Nightjar implements access control. The LLM is only allowed to lookup `<var>`-denoted variables and to write to `<:var>`-denoted variables. This means the LLM can only access data reachable through `<var>` and `<:var>` denoted variables. We did not implement type checking in our implementation, having observed that the system is performant even without it. We leave for future work to explore more deeply on different system design choices, such as enforcing type-checking with `<:var : type>` syntax, where only a value of `type` can be assigned to the variable `var`.
> >
> > > sandboxing
> >
> > We used sandboxing with Docker to perform our evaluation. We will release the code with the Docker image upon publication. We also also include warnings for LLM-based execution.
> >
> > > NL commands that try invalid operations
> >
> > In terms of natural code that try invalid operations, as described above, Nightjar programmatically guards against invalid variable accesses. Attempting invalid variable accesses result in the effect handlering resuming with an error message, and the LLM is expected to retry/fix its trajectory. Nightjar also programmatically guards against invalid program labels when writing to the control state by performing a basic static analysis of the program structure to determine valid labels. For example, `break` is only available to the LLM when the natural code does not reside in a loop context. We have added this clarification to the revised draft.

---

> > > ### Author Response · Authors · 2025-11-21
> > >
> > > > measuring how many errors occur, how they are detected and recovered
> > >
> > > We have incorporated a failure analysis in the revised draft and include it here for convenience. We analyzed the effect traces from SPSBench for Sonnet 4 and for GPT-4.1 and labeled the reason the LLM failed to get a perfect score on the program for each run of the program. Most program runs pass all test cases (`100% Pass`). It is only in a few instances that the LLM gives up on executing the natural code when it encounters an error (`Give Up`), hallucinates data (`Hallucination`), or incorrectly performed the program state operation specified in the natural code (`Incorrect State Op`). Most of the time, the LLM fails test cases due to making incorrect reasoning conclusions or math computations (`Reasoning`). Decreasing failures due to incorrect reasoning requires more capable LLM models or additional prompt engineering on the program's natural code or the LLM system prompt.
> > >
> > > | Category             | Sonnet 4 - Nightjar (Ours)   | GPT-4.1 - Nightjar (Ours)   |
> > > |----------------------|------------------------------|-----------------------------|
> > > | 100% Pass            | 82 (65.6%)                   | 69 (55.2%)                  |
> > > | Reasoning            | 41 (32.8%)                   | 47 (37.6%)                  |
> > > | Give Up              | 0 (0.0%)                     | 4 (3.2%)                    |
> > > | Failed State Op      | 1 (0.8%)                     | 5 (4.0%)                    |
> > > | Hallucination        | 1 (0.8%)                     | 0 (0.0%)                    |
> > > | Total                | 125                          | 125                         |
> > >
> > > We also further analyzed what kind of error messages the LLM encounters when executing natural code. These errors do not automatically result in execution failure; Nightjar gives the LLM these error messages and expects it to correct its mistake in executing the natural code. Sonnet 4 encountered very few errors, most of which are due to incorrect syntax or incorrect data type assumptions. GPT-4.1 encountered many more errors, most of them are due to confusing which effects to use (`State Op Error`). For example, it often tried to use `eval` instead of `exec`, before correcting its mistake to use `exec` successfully. It also often hallucinates variables (`Forbidden Var`) by confusing references for variable names or using `_` as a placeholder inappropriately. When encountering errors, the LLM generally corrects its mistake. GPT-4.1 sometimes ignores the error and performs a different operation instead or finds a workaround to performing the same computation.
> > >
> > > | Error Category       | Sonnet 4 - Nightjar (Ours)   | GPT-4.1 - Nightjar (Ours)   |
> > > |----------------------|------------------------------|-----------------------------|
> > > | State Op Error       | 0 (0.0%)                     | 219 (67.8%)                 |
> > > | Forbidden Var        | 0 (0.0%)                     | 70 (21.7%)                  |
> > > | Type Error           | 4 (36.4%)                    | 15 (4.6%)                   |
> > > | Syntax Error         | 4 (36.4%)                    | 13 (4.0%)                   |
> > > | Function Call Error  | 2 (18.2%)                    | 2 (0.6%)                    |
> > > | Index Error          | 1 (9.1%)                     | 4 (1.2%)                    |
> > > | Total                | 11                           | 323                         |
> > >
> > > | Recovery Method   | Sonnet 4 - Nightjar (Ours)   | GPT-4.1 - Nightjar (Ours)   |
> > > |-------------------|------------------------------|-----------------------------|
> > > | Correction        | 7 (63.6%)                    | 199 (61.6%)                 |
> > > | Ignored Error     | 1 (9.1%)                     | 59 (18.3%)                  |
> > > | Workaround        | 2 (18.2%)                    | 34 (10.5%)                  |
> > > | Retry             | 0 (0.0%)                     | 22 (6.8%)                   |
> > > | Not Recovered     | 1 (9.1%)                     | 9 (2.8%)                    |
> > >
> > > We have added this failure analysis to `Appendix M` of the revised draft. We have also added an example effect trace that did not pass a test case, an effect trace that passed the test case in `Appendix M`.

---

> > > > ### Author Response · Authors · 2025-11-21
> > > >
> > > > > tracking state invariants
> > > >
> > > > The SPSBench benchmarks already account for state invariants when evaluating the pass rate. For example, in `battle`:
> > > > ```
> > > > class Player:
> > > >     def __init__(self):
> > > >         self.name = "Knight Giar"
> > > >         self.health = 100
> > > >         self.attack_power = 10
> > > >
> > > > class Enemy:
> > > >     def __init__(self, name: str, health: int, attack_power: int):
> > > >         self.name = name
> > > >         self.health = health
> > > >         self.attack_power = attack_power
> > > >
> > > > def main(enemies: List[Enemy], action: str):
> > > >     player = Player()
> > > >     """natural
> > > >     Update each entity in <enemies> based on the <player>'s <action> and their stats.
> > > >     """
> > > >     return player, enemies
> > > > ```
> > > >
> > > > the evaluation checks whether only the specified enemy object is updated and whether the other enemy objects are unmodified.
> > > >
> > > > We acknowledge that Nightjar does not enable programmers to specify invariants that the LLM must maintain during execution, and that exploring different interfaces and additional verification and safety features for shared state programs is interesting future work.
> > > >
> > > > > Is it possible to add a discussion on how debugging mixed NL+code state will be supported (logging, visualization, stepping)?
> > > >
> > > > In terms of loggin and visualizing, we implemented a verbose parameter in Nightjar that stores/prints the emitted effects. For stepping, because Nightjar is implemented as a Python library, standard Python debugging tools can be used to set breakpoints and step over and into each effect. We anticipate future work on development tools and program analyses that assist programmers in writing natural code that achieves low execution time and high accuracy (Liang et al., 2025). Specialized testing tools in line with prompt testing (Sharma et al., 2025) could be also beneficial.
> > > >
> > > > We have incorporated a paragraph discussing future work on development tools in the Discussion & Conclusion section.
> > > >
> > > > ## References
> > > > [1] Omar Khattab, Arnav Singhvi, Paridhi Maheshwari, Zhiyuan Zhang, Keshav Santhanam, Sri Vardhamanan, Saiful Haq, Ashutosh Sharma, Thomas T. Joshi, Hanna Moazam, Heather Miller, Matei Zaharia, and Christopher Potts. Dspy: Compiling declarative language model calls into self-improving pipelines. In International Conference on Learning Representations, 2024.
> > > >
> > > > [2] Lianmin Zheng, Liangsheng Yin, Zhiqiang Xie, Chuyue Livia Sun, Jeff Huang, Cody Hao Yu, Shiyi Cao, Christos Kozyrakis, Ion Stoica, Joseph E Gonzalez, et al. Sglang: Efficient execution of structured language model programs. In Advances in Neural Information Processing Systems, 2024.
> > > >
> > > > [3] Karl Cobbe, Vineet Kosaraju, Mohammad Bavarian, Mark Chen, Heewoo Jun, Lukasz Kaiser, Matthias Plappert, Jerry Tworek, Jacob Hilton, Reiichiro Nakano, et al. Training verifiers to solve math word problems. arXiv preprint arXiv:2110.14168, 2021.
> > > >
> > > > [4] David Rein, Betty Li Hou, Asa Cooper Stickland, Jackson Petty, Richard Yuanzhe Pang, Julien Dirani, Julian Michael, and Samuel R Bowman. Gpqa: A graduate-level google-proof q&a benchmark. In First Conference on Language Modeling, 2024.
> > > >
> > > > [5] Zhilin Yang, Peng Qi, Saizheng Zhang, Yoshua Bengio, William Cohen, Ruslan Salakhutdinov,
> > > > and Christopher D Manning. Hotpotqa: A dataset for diverse, explainable multi-hop question
> > > > answering. In Proceedings of the 2018 conference on empirical methods in natural language
> > > > processing, pp. 2369–2380, 2018.
> > > >
> > > > [6] François Chollet. On the measure of intelligence. arXiv preprint arXiv:1911.01547, 2019.

---

### Official Review · Reviewer_83Tw · 2025-11-01

**Soundness:** 3
**Presentation:** 4
**Contribution:** 3
**Rating:** 6
**Confidence:** 4

**Summary:**

The authors propose to enable LLM "execution" agents to directly interact with a host language program state to facilitate easier interaction between "natural code" and formal language code (programming languages). They define a Natural Function Interface (NFI) along with its operational semantics, and show how NFIs generalise tool use/MCP to a richer interaction semantics (shared state, shared control flow).
The authors then curate a benchmark dataset with natural code to evaluate their approach in terms of lines of code (after normalisation), pass-rate, and execution time. They show a significant reduction of code size (by making much of the serialisation, reisation implicit to the paradigm) while matching the manually written implementations in a majority of cases.

**Strengths:**

- Generalisation of tool use to shared program state (memory, control)
- competitive results even with what seems like a naive implementation

**Weaknesses:**

- It is unclear how/where serialisation/data marshalling is to be implemented; it is also unclear if grammar-based sampling is compatible with the approach or if any such grammar file should be augmented to enable emission of effect tokens.
- perhaps out of scope, but the handler loop seems unintuitive: if emitted effects are side-effect free wrt the program state that is observed by the NL code, why not stage an execution plan and delay the interrupts until the last moment possible to reduce overhead?
- perhaps missing reference to literate programming (of ye olden days): Knuth, Donald Ervin. "Literate programming." The computer journal 27.2 (1984): 97-111. Not directly the same concept, but it follows the spirit of literate programming
- There is an unexplored trade-off frontier: in the paper, the comparison is the paradigm handling the boilerplate vs the user-written boilerplate. A fully FL/PL program vs a hybrid program would be another comparison of interest, but the cost normalisation is more difficult.

**Questions:**

- Q1: The idea of shared state requires the model to be aware of how data can be manipulated (`graph.add_edge((14,5))`, etc.) and moving data via the handler requires serialisation and conversion back to the expected data-type (similar to FFI dtypes vs the host language dtypes). Who has the burden of defining these serialisation operations, Function symbol discovery, etc.?

- Q2: While the implicit to the paradigm operations vs explicit boilerplate is explored in the paper, how does this compare to generating a single PL script that solves the task instead of switching PL/NL context?

- Q3: How does NL execution handle stochasticity? Is the temperature set to 0?

---

> ### Author Response · Authors · 2025-11-21
>
> We thank the reviewer for the thoughtful comments and questions. We address each of them below:
>
> > It is unclear how/where serialisation/data marshalling is to be implemented
>
> Serialization/data marshaling is implemented by the shared program state system developer, not the program developer. Specifically, the system developer must implement a serialization function $\alpha: v \rightarrow v_N$ and reification function $\gamma: v_N \rightarrow v$ that Nightjar calls on the values to be resumed and the parameters of the effects. Concretely, the formalism is as follows:
> ```
> handle_N e_N with {
>     finally v_N -> γ(v_N);
>     Lookup(x) -> resume α(x);
>     Assign(x, v_N) -> let x = γ(v_N) in resume α(());
>     Deref(a) -> resume α(!γ(a));
>     Ref(v_N) -> resume α(ref γ(v_N));
>     Labels() -> resume α(labels);
>     Goto(l, v_N) -> goto γ(l) γ(v_N);
> }
> ```
> where $v_N$ are NFI values and $v$ are host values. We have incorporated this clarification to the NFI in the revised draft.
>
> > it is also unclear if grammar-based sampling is compatible with the approach or if any such grammar file should be augmented to enable emission of effect tokens.
>
> Assuming the reviewer is asking whether grammar-based structured output by the LLM is compatible with generating effects during program execution, we make the following clarification: Our implementation uses JSON format to encode the effects in order to enforce structured output, but other LLM response formats are also viable design choices that the system developer may choose. We have supplied a straightfoward implementation in the paper to demonstrate the efficacy of enabling executable shared program state programs.
>
> We kindly request the reviewer to clarify their question if we have not addressed it.
>
> > perhaps out of scope, but the handler loop seems unintuitive: if emitted effects are side-effect free wrt the program state that is observed by the NL code, why not stage an execution plan and delay the interrupts until the last moment possible to reduce overhead?
>
> The handler presented in Figure 3 is a formal specification of the semantics of the interface. Nightjar's implementation with an LLM agent loop is one implementation of the formal specification. There is a tradeoff between an interpreter approach (i.e. Nightjar) and a just-in-time (JIT) compiled (i.e. staged execution plan) approach. In the interpreted approach, the LLM can adapt the next step to the result of the previous effects. Even side-effect free ones that only read values from the program state can change the trajectory of the following effects. The cost to this adaptibility is runtime overhead. In the JIT-compiled approach, the runtime overhead of LLM roundtrips would be reduced, but the LLM cannot adapt to intermediate program values. The goal of this work is to present the shared program state abstraction and demonstrate that it can be realized as executable programs. We leave deeper exploration of different execution and compilation strategies for future work.
>
> > perhaps missing reference to literate programming (of ye olden days): Knuth, Donald Ervin. "Literate programming." The computer journal 27.2 (1984): 97-111.
>
> We have incorporated Literate Programming to Related Work.
>
> > There is an unexplored trade-off frontier: in the paper, the comparison is the paradigm handling the boilerplate vs the user-written boilerplate. A fully FL/PL program vs a hybrid program would be another comparison of interest, but the cost normalisation is more difficult.

---

> > ### Author Response · Authors · 2025-11-21
> >
> > > Q2: While the implicit to the paradigm operations vs explicit boilerplate is explored in the paper, how does this compare to generating a single PL script that solves the task instead of switching PL/NL context?
> >
> > All the programs in SPSBench require use of LLM inference at runtime (e.g. natural language processing, sentiment analysis). Therefore, LLM-free implementations are not appropriate for these tasks. In the event that the reviewer is instead referring to using an LLM to statically generate Python code (that can include LLM API calls) given the source code program, we have implemented and evaluated this method of executing shared state programs (`LLM Compiler`). We show below the average pass rate with standard deviation and average times across 5 runs. (See the revised paper for the time ranges).
> >
> > | Model    | Method                      | Pass Rate | Runtime | Compile Time | Total Time |
> > |----------|-----------------------------|-----------|---------|--------------|----------- |
> > | Sonnet 4 | Programmer Impl             | 0.78±0.03 | 8.0   | -            | 8.0       |
> > | Sonnet 4 | LLM Compiler                | 0.36±0.06 | 7.9    | 7.3         | 15.2      |
> > | Sonnet 4 | Nightjar (Ours)             | 0.85±0.02 | 75.8   | 0.0         | 75.8      |
> > | GPT-4.1  | Programmer Impl             | 0.74±0.03 | 4.5    | -            | 4.5       |
> > | GPT-4.1  | LLM Compiler                | 0.45±0.02 | 2.8    | 5.0         | 7.7       |
> > | GPT-4.1  | Nightjar (Ours)             | 0.81±0.04 | 71.8   | 0.0         | 71.8      |
> >
> > *Note: We corrected bugs in `balanced_parenthesis`, `calendar`, `coffeeshop`, `email_classification`, and `stores` that improved pass rates for all techniques versus the original paper, but still preserved the conclusions. We have updated the result table in the revised draft accordingly.
> >
> > As shown above, using LLM to generate Python code performs poorly in accuracy, as the LLM has no visibility to program inputs and cannot adapt accordingly. We have included the additional results in `Appendix H` in the revised draft.
> >
> > > Q1: [...] Who has the burden of defining these serialisation operations, Function symbol discovery, etc.?
> >
> > This is an aspect that is completely hidden from the program developer. The system developer of the shared program state programming system has the burden of defining serialization/reification.
> >
> > Nightjar only enables lookups for `<var>`-denoted names and assigns for `<:var>`-denoted names by the LLM interpreter as a safety precaution. In Python, function names are in the same space as variable names. As such, the LLM interpreter only has access to using functions that are `<var>`-denoted in the natural blocks or custom functions it writes itself.
> >
> > > Q3: How does NL execution handle stochasticity? Is the temperature set to 0?
> >
> > In our evaluation, the temperature is set to 1. As such, the Nightjar executions may differ in the effect trace at every run. If this stochasticity is not desired by the program developer or system developer, using temperature 0 on a open-source LLM model or caching the responses to the same effect trace will remove this stochasticity in executions.

---

> > > ### Comment · Reviewer_83Tw · 2025-11-21
> > >
> > > Thank you for the answers and clarifications.
> > >
> > > Regarding grammar output, I did indeed mean enforcing structured output according to, for example, a gbnf grammar file. The use of JSON for encoding effects suggests that that could be compatible if the desired grammar is embedded into a JSON string or by other similar means. This addresses my question. I echo back a similar request; if I misunderstood, I kindly ask that I be corrected.
> > >
> > > For Q1/Q3, thank you for the answers. Regarding Q3, I feel this design dimension should be communicated. In an execution setting/developer case-study, I understand the choice of t=1, but under reproducability, this complicates the situation. Regardless, I am satisfied with the clarification.
> > >
> > > For Q2, I did mean having the LLM produce the script (which may admit LLM calls). So the LLM Compile setting answers the question. Thank you.

---

> > > > ### Author Response · Authors · 2025-11-25
> > > >
> > > > > Regarding grammar output, I did indeed mean enforcing structured output according to, for example, a gbnf grammar file. The use of JSON for encoding effects suggests that that could be compatible if the desired grammar is embedded into a JSON string or by other similar means. This addresses my question. I echo back a similar request; if I misunderstood, I kindly ask that I be corrected.
> > > >
> > > > We thank the reviewer for the clarification. That is correct: using a grammar-based structured output to encode effects is a viable alternative implementation.
> > > >
> > > > > Regarding Q3, I feel this design dimension should be communicated. In an execution setting/developer case-study, I understand the choice of t=1, but under reproducability, this complicates the situation.
> > > >
> > > > We have updated the revised draft with the clarification and implications of the temperature setting and added the following results to `Appendix N`. With respect to reproducibility, because we used closed LLMs as the base LLM, the results might not be deterministic even with t=0 because of hardware routing, GPU non-determinism, batch invariance etc.[1,2].
> > > >
> > > > To illustrate that even with t=0, the LLM responses are non-deterministic, we ran the SPSBench evaluation with t=0 on GPT-4.1 with 5 runs. Even with t=0, there is still a variance in the pass rate, indicating the responses are non-deterministic. We also note that the same performance trends from t=1 are exhibited here.
> > > >
> > > > | Model       | Method                      | Pass Rate   | Runtime (s)    | # Effects |
> > > > |-------------|-----------------------------|-------------|----------------|----------------|
> > > > | GPT-4.1     | Programmer Impl             | 0.74±0.01   | 5.0            | -   |
> > > > | GPT-4.1     | Nightjar (Ours)             | 0.73±0.03   | 82.9           | 79.9  |
> > > >
> > > > [1] https://platform.claude.com/docs/en/api/kotlin/completions/create
> > > >
> > > > [2] He, Horace and Thinking Machines Lab, “Defeating Nondeterminism in LLM Inference”, Thinking Machines Lab: Connectionism, Sep 2025.

---

> > > > > ### Comment · Reviewer_83Tw · 2025-11-26
> > > > >
> > > > > Thank you for the follow-up, and fair point regarding non-determinism on commercial platforms due to additional factors. Communicating this aspect is sufficient to address my concern on this matter.
> > > > >
> > > > > I maintain my positive score on this work.

---

### Official Review · Reviewer_M1qm · 2025-11-02

**Soundness:** 3
**Presentation:** 3
**Contribution:** 3
**Rating:** 8
**Confidence:** 4

**Summary:**

This paper introduces shared programs states between formal and natural programming languages. This is formalized with the concept of Natural Function Interfaces (NFIs), through values, effects and handlers for sharing scopes, heaps, and control states. The proposed system, NIGHTJAR, implements this shared program and enables natural language code to directly read, write, and manipulate the program state of a host language such as Python. The work is supported by theoretical grounding and a new benchmark suite (SPSBench) evaluating program conciseness and performance.

**Strengths:**

- **Novel conceptual contribution**: The idea of exposing host-language state and control to an LLM in a principled and programmable way is novel. This moves beyond existing systems that treat LLMs as isolated components that make tool calls using programmer defined functions.
- **Strong formal framework**: The authors present a formal framework for NFIs, including variable scopes, heap references, and control state.
- **System implementation**: The NIGHTJAR system demonstrates that the abstraction is realizable and yields substantial code reduction  compared to manually writing interoperable code.

**Weaknesses:**

- **Limited empirical evaluation**: The evaluation on program pass rates and conciseness, while adequate as proof of concept, does not deeply explore scalability, robustness, or user experience.
- **Benchmarks are synthetic**: SPSBench appears to consist mainly of small programs adapted from documentation examples. It would be great to have analyses on real-world user code or larger-scale applications.
- **Safety and correctness**: Though acknowledged in discussion, the implications of allowing natural code direct memory access (even through abstraction) deserve stronger treatment.

**Questions:**

- **Execution times may be conservative**: Rather than relying only on LLM APIs, consider adding evaluations with smaller, locally hosted LLMs.
- **Discuss failure modes**: Consider adding some analysis of where failures occur when they do.

---

> ### Author Response · Authors · 2025-11-21
>
> We thank the reviewer for the thoughtful comments and questions. We address each of them below:
>
> > scalability, robustness
>
> We have performed an empirical evaluation on the scalability of using shared program state with large data structures, which we included in `Appendix F.3`. We compared the performance of the Figure 1a) program and Figure 1c) program as we increased the graph sizes from 25 nodes to 5000 nodes. The results showed that Nightjar is performant on large graphs, because enabling shared heap enables the LLM to only access the parts of the graph required to answer user queries. In comparison, the programmer implementation (which serializes the whole graph and reifies the whole output graph) fails to perform on graphs larger than 1000 nodes, as the encoded graph exceeds the context window. While the programmer can manually implement workarounds in the program to avoid serializing graphs, Nightjar does not need any additional changes to the Figure 1a) program to facilitate adapting computation to program inputs and resources.
>
> > user experience
>
> Our evaluation strategy is consistent with prior published works in ICLR and NeuRIPS related to enabling interoperability between natural code and formal programs such as DSPy[1] and SGLang[2]. Moreover, in addition to publication, both of these systems have seen significant public adoption since publication and neither performs a user study. These papers instead focus on the technical underpinnings of the programming model, as we do in our paper. As such, a complete user study on the many different frameworks for embedding natural code in formal programs is beyond the scope of this work.
>
> > Benchmarks are synthetic: [...] analyses on real-world user code or larger-scale applications.
>
> The sizes of the SPSBench programs are also comparable to the size of programs evaluated in prior published works like DSPy[1] and SGLang[2].
>
> > Safety and correctness: Though acknowledged in discussion, the implications of allowing natural code direct memory access (even through abstraction) deserve stronger treatment.
>
> As described in Section 5, Nightjar implements access control as safety precautions. The LLM is only allowed to lookup `<var>`-denoted variables and to write to `<:var>`-denoted variables. This means the LLM can only access data reachable through `<var>` and `<:var>` denoted variables. On the SPSBench programs, we found this level of access to be adequate.
>
> > consider adding evaluations with smaller, locally hosted LLMs.
>
> We repeated the experiments on SPSBench using GPT-OSS 20B, served using vLLM on a machine with an A100 80GB GPU. The accuracy of all techniques drop with the smaller GPT-OSS 20B model being less capable than the closed models. In contrast to our results with closed models, the programming implementation is more accurate than Nightjar. Nightjar requires a base LLM to have strong agentic capabilities. Nonetheless, Nightjar is still more accurate than using static code generation (LLM Compiler). In LLM Compiler, we query the LLM to statically generate Python code (that can contain LLM API calls) to replace the natural block given the whole program. Below, we show the average pass rate with standard deviation, average runtime, compile time, total time, and average number of emitted effects. We have added these results to `Appendix J` of the revised draft.
>
> | Model       | Method                      | Pass Rate | Runtime | Compile Time | Total Time | # Effects    |
> |-------------|-----------------------------|-----------|---------|--------------|------------|------------|
> | GPT-OSS 20B | Programmer Impl             | 0.68±0.04 | 5.3    | -            | 5.3       | -          |
> | GPT-OSS 20B | LLM Compiler                | 0.13±0.03 | 1.2    | 4.3         | 5.3       | -          |
> | GPT-OSS 20B | Nightjar (Ours)             | 0.47±0.03 | 54.2   | 0.0         | 54.2      | 28.1      |

---

> > ### Author Response · Authors · 2025-11-21
> >
> > > Discuss failure modes: Consider adding some analysis of where failures occur when they do.
> >
> > We have incorporated a failure analysis in the revised draft and include it here for convenience. We analyzed the effect traces from SPSBench for Sonnet 4 and for GPT-4.1 and labeled the reason the LLM failed to get a perfect score on the program for each run of the program. Most program runs pass all test cases (`100% Pass`). It is only in a few instances that the LLM gives up on executing the natural code when it encounters an error (`Give Up`), hallucinates data (`Hallucination`), or incorrectly performed the program state operation specified in the natural code (`Incorrect State Op`). Most of the time, the LLM fails test cases due to making incorrect reasoning conclusions or math computations (`Reasoning`). Decreasing failures due to incorrect reasoning requires more capable LLM models or additional prompt engineering on the program's natural code or the LLM system prompt.
> >
> > | Category             | Sonnet 4 - Nightjar (Ours)   | GPT-4.1 - Nightjar (Ours)   |
> > |----------------------|------------------------------|-----------------------------|
> > | 100% Pass            | 82 (65.6%)                   | 69 (55.2%)                  |
> > | Reasoning            | 41 (32.8%)                   | 47 (37.6%)                  |
> > | Give Up              | 0 (0.0%)                     | 4 (3.2%)                    |
> > | Incorrect State Op   | 1 (0.8%)                     | 5 (4.0%)                    |
> > | Hallucination        | 1 (0.8%)                     | 0 (0.0%)                    |
> > | Total                | 125                          | 125                         |
> >
> > We also further analyzed the types of error messages the LLM encounters when issuing effects and how it attempts to recover from these errors. Sonnet 4 encountered very few errors, most of which are due to incorrect syntax or incorrect data type assumptions. GPT-4.1 encountered many more errors. Most of them are due to confusing which effects to use (`State Op Error`). For example, it often tried to use `eval` instead of `exec`, before correcting its mistake to use `exec` successfully. It also often hallucinates variables (`Forbidden Var`) by confusing references for variable names or using `_` as a placeholder inappropriately. When encountering errors, the LLM generally corrects its mistake. GPT-4.1 sometimes ignores the error and performs a different operation instead or finds a workaround to performing the same computation.
> >
> > | Error Category       | Sonnet 4 - Nightjar (Ours)   | GPT-4.1 - Nightjar (Ours)   |
> > |----------------------|------------------------------|-----------------------------|
> > | State Op Error       | 0 (0.0%)                     | 219 (67.8%)                 |
> > | Forbidden Var        | 0 (0.0%)                     | 70 (21.7%)                  |
> > | Type Error           | 4 (36.4%)                    | 15 (4.6%)                   |
> > | Syntax Error         | 4 (36.4%)                    | 13 (4.0%)                   |
> > | Function Call Error  | 2 (18.2%)                    | 2 (0.6%)                    |
> > | Index Error          | 1 (9.1%)                     | 4 (1.2%)                    |
> > | Total                | 11                           | 323                         |
> >
> > | Recovery Method   | Sonnet 4 - Nightjar (Ours)   | GPT-4.1 - Nightjar (Ours)   |
> > |-------------------|------------------------------|-----------------------------|
> > | Correction        | 7 (63.6%)                    | 199 (61.6%)                 |
> > | Ignored Error     | 1 (9.1%)                     | 59 (18.3%)                  |
> > | Workaround        | 2 (18.2%)                    | 34 (10.5%)                  |
> > | Retry             | 0 (0.0%)                     | 22 (6.8%)                   |
> > | Not Recovered     | 1 (9.1%)                     | 9 (2.8%)                    |
> >
> > We have added this failure analysis to `Appendix M` of the revised draft. We have also added an example effect trace that did not pass a test case, an effect trace that passed the test case in `Appendix M`.
> >
> > ## References
> >
> > [1] Omar Khattab, Arnav Singhvi, Paridhi Maheshwari, Zhiyuan Zhang, Keshav Santhanam, Sri Vardhamanan, Saiful Haq, Ashutosh Sharma, Thomas T. Joshi, Hanna Moazam, Heather Miller, Matei Zaharia, and Christopher Potts. Dspy: Compiling declarative language model calls into self-improving pipelines. In International Conference on Learning Representations, 2024.
> >
> > [2] Lianmin Zheng, Liangsheng Yin, Zhiqiang Xie, Chuyue Livia Sun, Jeff Huang, Cody Hao Yu, Shiyi Cao, Christos Kozyrakis, Ion Stoica, Joseph E Gonzalez, et al. Sglang: Efficient execution of structured language model programs. In Advances in Neural Information Processing Systems, 2024.

---

### Official Review · Reviewer_8swZ · 2025-11-04

**Soundness:** 2
**Presentation:** 3
**Contribution:** 1
**Rating:** 2
**Confidence:** 3

**Summary:**

This paper introduces the idea of shared program state, essentially an LLM black box that can read and manipulate the program's internal state which can be steered somewhat by specialised syntax. The paper walks this idea through with an example, defines the proposed construct in eBNF and further explains the functionality of each component. The paper concludes with a reference implementation, nightjar, of the proposed idea and a lightweight empirical study with 25 programming examples which suggests that programmers using nightjar are about as proficient as programmers without it, while sacrificing execution time for conciseness.

**Strengths:**

I think the general idea of natural language programming interfaces is intriguing and has a rich history that goes well beyond the current LLM hype - indeed beyond deep learning.

**Weaknesses:**

I am not convinced ICLR is the best avenue to publish this work. The empirical study is at best preliminary and I am not convinced at all that programmers would like to give up control and lose out on execution time for saving a few extra lines of code - with LLM-assisted code writing and analysis tools these come essentially for free.

I suggest to greatly expand the scope of the empirical analysis, include evaluations on known public benchmarks as well as to incorporate qualitative feedback, analysis of failure modes, etc, to increase the scope of the contribution commesurate with ICLR.

**Questions:**

No questions at this point.

---

> ### Author Response · Authors · 2025-11-21
>
> We thank the reviewer for the thoughtful comments. We address each of them below:
>
> > I am not convinced at all that programmers would like to give up control and lose out on execution time for saving a few extra lines of code - with LLM-assisted code writing and analysis tools these come essentially for free.
>
> LLM-assisted code writing and analysis tools are only capable of adding fixed. code to the program, the content of which cannot adapt dynamically to program inputs (given in natural language or otherwise) or to intermediate program values. While fixed code generation is still a powerful capability, we have designed to generalize to computations that require more than fixed code generation.
>
> To demonstrate this, we evaluated the performance of a compiler-based approach to execute shared program state programs. In this approach, we query the LLM to statically generate Python code (that can make LLM API calls) to replace the natural block given the whole program source code. This approach produces only 36% pass rate with Sonnet 4 and 45% with GPT-4.1. We show the average pass rate with standard deviation and average runtime, compile time, and total time across 5 runs. (See `Appendix H` in the revised paper for the time ranges).
>
> | Model    | Method                      | Pass Rate | Runtime | Compile Time | Total Time |
> |----------|-----------------------------|-----------|---------|--------------|----------- |
> | Sonnet 4 | Programmer Impl             | 0.78±0.03 | 8.0    | -            | 8.0       |
> | Sonnet 4 | LLM Compiler                | 0.36±0.06 | 7.9    | 7.3         | 15.2      |
> | Sonnet 4 | Nightjar (Ours)             | 0.85±0.02 | 75.8   | 0.0         | 75.8      |
> | GPT-4.1  | Programmer Impl             | 0.74±0.03 | 4.5    | -            | 4.5       |
> | GPT-4.1  | LLM Compiler                | 0.45±0.02 | 2.8    | 5.0         | 7.7       |
> | GPT-4.1  | Nightjar (Ours)             | 0.81±0.04 | 71.8   | 0.0         | 71.8      |
>
> *Note: We corrected bugs in `balanced_parenthesis`, `calendar`, `coffeeshop`, `email_classification`, and `stores` that improved pass rates for all techniques versus the original paper, but still preserved the conclusions. We have updated the result table in the revised draft accordingly.
>
> To utilize the convenience of writing high-level prompts while still maintaining accuracy to manually written low-level code, we believe shared program state provides a valueable and viable mode of programming.
>
> We further demonstrate that the overhead of using shared program state can be reduced with additional engineering on the Nightjar system. We implemented a version of Nightjar in which the system eagerly gives the type, heap references, and values of `<var>`-denoted variables to the LLM along with the natural code to execute. We refer to this version as `Optimized`. The optimization maintains the pass rate and reduces the runtime by 2.9-4.0x from the vanilla Nightjar implementation. Specifically, the optimization reduces the overhead of waiting for the LLM to emit effects to read the program state, since it is almost always the case that the LLM should read `<var>`-denoted variables in the natural code. This is shown by the reduced average number of effects used by the Optimized Nightjar. We have included these results in the Evaluation section of the revised draft.
>
> | Model    | Method                      | Pass Rate | Runtime | # Effects |
> |----------|-----------------------------|-----------|---------|---|
> | Sonnet 4 | Programmer Impl             | 0.78±0.03 | 8.0    | -|
> | Sonnet 4 | Nightjar (Ours)             | 0.85±0.02 | 75.8   | 27.3|
> | Sonnet 4 | Nightjar (Ours) (Optimized) | 0.85±0.03 | 25.9   | 8.9 |
> | GPT-4.1  | Programmer Impl             | 0.74±0.03 | 4.5    | -|
> | GPT-4.1  | Nightjar (Ours)             | 0.81±0.04 | 71.8   | 63.1 |
> | GPT-4.1  | Nightjar (Ours) (Optimized) | 0.78±0.03 | 19.6   | 16.6 |

---

> > ### Author Response · Authors · 2025-11-21
> >
> > > include evaluations on known public benchmarks
> >
> > While programmers are increasingly integrating prompts into programs, no existing benchmarks that we know of tests the interoperability of programs with embedded natural code. Existing LLM benchmarks such as GSM8K[1], GPQA[2], HotPotQA[3], ARC[4] test the inherent reasoning and problem solving capabilities of LLMs; they are not designed to test the performance of a program that utilizes both formal program constructs/data structures and natural code.
> >
> > As these existing benchmarks do not interact with formal program state, they do not test the capability of an LLM system to perform operations on the program state. This is why we developed the SPSBench benchmark suite.
> >
> > Upon publication, we will release SPSBench as a public benchmark for evaluating systems that enable shared program state.
> >
> > For example, a test problem from GSM8K is
> > ```
> > A robe takes 2 bolts of blue fiber and half that much white fiber. How many bolts in total does it take?
> > ```
> > Notably, this test problem is entirely natural code and does not meaningfully utilize a formal program state, because GSM8K is fundamentally a benchmark for evaluating an LLM's math reasoning capability, not natural-formal code interoperability.
> >
> > We have further evaluated on all 1319 problems from the GSM8K eval set using GPT-4.1 as the LLM. We compare against the programmer implementation program, which queries the LLM for a string expression that it invokes the Python `eval` function on. We also compare against a version of programmer implementation that uses tool use with the OpenAI Code Interpreter (which evaluates code in an isolated container) and a version that implements tool use with a custom, user-defined code evaluation tool that maintain isolated program state.
> >
> > We show the average pass rate, runtime, and number of tool calls or effects over 3 runs. As shown, Nightjar achieves higher or equal accuracy to the programmer implementations, demonstrating that the setup does not detract from LLM reasoning capabilities. Nightjar incurs slight runtime overhead due to emitting effects, but 1) the runtime is approximately the same as using OpenAI's code interpreter tool in the programmer implementation and 2) the overhead can be reduced with engineering efforts as shown by Optimized Nightjar. We have added these results to `Appendix L` of the revised draft.
> >
> > | Method                      | Pass Rate | Runtime | # Tool Calls/Effects |
> > |-----------------------------|-----------|---------|----------------------|
> > | GPT-4.1  | Programmer Impl                    | 0.84±0.01   | 1.4    | 0.0     |
> > | GPT-4.1  | Programmer Impl  w/ Code Interpreter  | 0.92±0.01   | 5.1   | 0.9     |
> > | GPT-4.1  | Programmer Impl w/  Custom Code Interpreter | 0.70±0.01   | 2.6    | 1.6     |
> > | GPT-4.1  | Nightjar (Ours)                    | 0.93±0.00   | 5.9    | 6.6    |
> > | GPT-4.1  | Nightjar (Ours) (Optimized)        | 0.93±0.00   | 4.7    | 4.1    |

---

> > > ### Author Response · Authors · 2025-11-21
> > >
> > > > as well as to incorporate qualitative feedback
> > > > analysis of failure modes
> > >
> > > We have incorporated a failure analysis in the revised draft and include it here for convenience. We analyzed the effect traces from SPSBench for Sonnet 4 and for GPT-4.1 and labeled the reason the LLM failed to get a perfect score on the program for each run of the program. Most program runs pass all test cases (`100% Pass`). It is only in a few instances that the LLM gives up on executing the natural code when it encounters an error (`Give Up`), hallucinates data (`Hallucination`), or incorrectly performed the program state operation specified in the natural code (`Incorrect State Op`). Most of the time, the LLM fails test cases due to making incorrect reasoning conclusions or math computations (`Reasoning`). Decreasing failures due to incorrect reasoning requires more capable LLM models or additional prompt engineering on the program's natural code or the LLM system prompt.
> > >
> > > | Category             | Sonnet 4 - Nightjar (Ours)   | GPT-4.1 - Nightjar (Ours)   |
> > > |----------------------|------------------------------|-----------------------------|
> > > | 100% Pass            | 82 (65.6%)                   | 69 (55.2%)                  |
> > > | Reasoning            | 41 (32.8%)                   | 47 (37.6%)                  |
> > > | Give Up              | 0 (0.0%)                     | 4 (3.2%)                    |
> > > | Incorrect State Op   | 1 (0.8%)                     | 5 (4.0%)                    |
> > > | Hallucination        | 1 (0.8%)                     | 0 (0.0%)                    |
> > > | Total                | 125                          | 125                         |
> > >
> > > We also further analyzed the types of error messages the LLM encounters when issuing effects and how it attempts to recover from these errors. Sonnet 4 encountered very few errors, most of which are due to incorrect syntax or incorrect data type assumptions. GPT-4.1 encountered many more errors. Most of them are due to confusing which effects to use (`State Op Error`). For example, it often tried to use `eval` instead of `exec`, before correcting its mistake to use `exec` successfully. It also often hallucinates variables (`Forbidden Var`) by confusing references for variable names or using `_` as a placeholder inappropriately. When encountering errors, the LLM generally corrects its mistake. GPT-4.1 sometimes ignores the error and performs a different operation instead or finds a workaround to performing the same computation.
> > >
> > > | Error Category       | Sonnet 4 - Nightjar (Ours)   | GPT-4.1 - Nightjar (Ours)   |
> > > |----------------------|------------------------------|-----------------------------|
> > > | State Op Error       | 0 (0.0%)                     | 219 (67.8%)                 |
> > > | Forbidden Var        | 0 (0.0%)                     | 70 (21.7%)                  |
> > > | Type Error           | 4 (36.4%)                    | 15 (4.6%)                   |
> > > | Syntax Error         | 4 (36.4%)                    | 13 (4.0%)                   |
> > > | Function Call Error  | 2 (18.2%)                    | 2 (0.6%)                    |
> > > | Index Error          | 1 (9.1%)                     | 4 (1.2%)                    |
> > > | Total                | 11                           | 323                         |
> > >
> > > | Recovery Method   | Sonnet 4 - Nightjar (Ours)   | GPT-4.1 - Nightjar (Ours)   |
> > > |-------------------|------------------------------|-----------------------------|
> > > | Correction        | 7 (63.6%)                    | 199 (61.6%)                 |
> > > | Ignored Error     | 1 (9.1%)                     | 59 (18.3%)                  |
> > > | Workaround        | 2 (18.2%)                    | 34 (10.5%)                  |
> > > | Retry             | 0 (0.0%)                     | 22 (6.8%)                   |
> > > | Not Recovered     | 1 (9.1%)                     | 9 (2.8%)                    |
> > >
> > > We have added this failure analysis to `Appendix M` of the revised draft. We have also added an example effect trace that did not pass a test case, an effect trace that passed the test case in `Appendix M`.
> > >
> > > ## References
> > > [1] Karl Cobbe, Vineet Kosaraju, Mohammad Bavarian, Mark Chen, Heewoo Jun, Lukasz Kaiser, Matthias Plappert, Jerry Tworek, Jacob Hilton, Reiichiro Nakano, et al. Training verifiers to solve math word problems. arXiv preprint arXiv:2110.14168, 2021.
> > >
> > > [2] David Rein, Betty Li Hou, Asa Cooper Stickland, Jackson Petty, Richard Yuanzhe Pang, Julien Dirani, Julian Michael, and Samuel R Bowman. Gpqa: A graduate-level google-proof q&a benchmark. In First Conference on Language Modeling, 2024.
> > >
> > > [3] Zhilin Yang, Peng Qi, Saizheng Zhang, Yoshua Bengio, William Cohen, Ruslan Salakhutdinov, and Christopher D Manning. Hotpotqa: A dataset for diverse, explainable multi-hop question
> > > answering. In Proceedings of the 2018 conference on empirical methods in natural language processing, pp. 2369–2380, 2018.
> > >
> > > [4] François Chollet. On the measure of intelligence. arXiv preprint arXiv:1911.01547, 2019.

---

### Author Response · Authors · 2025-11-21
**Summary of Changes**

We thank the reviewers for the thoughtful feedback and questions. We have added additional discussions and results requested by reviewers to the revised draft highlighted in blue. We list below a summary of the changes:

- Related Work Section: Added paragraph headings and elaborated on differences between prior work and our work. Added related work mentioned by reviewers (Literate Programming, CoPrompt, ANPL, Prompting is Programming Too, Program of Thoughts Prompting)
- Shared Program State NFI Section: Added clarification to NFI on when serialization/reification happens
- Evaluation Section: Updated main experiment results and added Optimized Nightjar results
- Discussion & Conclusion: Added paragraph on future work on program development tool
- Appendix:
    - Added LLM Compiler results to Appendix H
    - Added Optimized Nightjar and caching results to Appendix I
    - Added results with GPT-OSS 20B to Appendix J
    - Added plot of the number of effects to runtime in Appendix K
    - Added GSM8K results to Appendix L
    - Added failure analysis to Appendix M

---

### Author Response · Authors · 2025-12-03
**Discussion Summary**

Dear Area Chair,

We thank you for your time and understand that the unprecedented circumstances may be difficult. We summarize the discussion thus far.

Reviewer 83Tw has confirmed we have addressed all their concerns and maintain their positive score.

We have also responded to the concerns raised by Reviewers 8swZ, M1qm, and KmWC. Specifically:
- **Failure Analysis & Additional Evaluation**: Reviewer 8swZ, M1qm, and KmWC suggested adding failure analysis, which we have added. Reviewer 8swZ and KmWC also suggested adding evaluations on existing benchmarks. Accordingly, we have added additional evaluation on GSM8K. Reviewer M1qm suggested evaluation on smaller, locally hosted LLMs, which we have added.
- **Runtime Overhead**: Reviewer 8swZ and KmWC has raised concerns about the runtime overhead of our Nightjar system compared to LLM-based code generation. We have added additional results that show LLM-based code generation performs poorly in terms of accuracy; as such, Nightjar’s runtime overhead represents a tradeoff for better accuracy (Nightjar achieves 81-85% accuracy compared to LLM-based code generation's 36-45% accuracy). This ablation was also requested by Reviewer 83Tw. We also added an optimized version of Nightjar which reduced the vanilla Nightjar's runtime by 2.9-3.7x, showing that the runtime overhead is reducible with additional engineering.
- **Related Work**: We have added the related works mentioned by Reviewer 83Tw and KmWC.
- **Access Control and Safety**: Reviewer M1qm and KmWC raised concerns about needing access control on the LLM interpreter of Nightjar. We have clarified that Nightjar does implement access control, in that the Nightjar system only allows the LLM to access `<var>` and `<:var>`-denoted variables and to jump to program labels that are valid in the evaluation context. This text is in Section 5 of the draft.

We thank the reviewers for their time and detailed feedback.

---

### Meta-Review · Area_Chair_83Xe · 2026-01-01

**Summary:**

he paper introduces Nightjar, a system implementing a Shared Program State abstraction that enables natural language prompts to directly access and manipulate a host program's variables, heap, and control flow, achieving comparable accuracy to manual code while significantly reducing lines of code (approx. 40%).

**Reviewer Concerns:**

The reviewers' main outstanding concerns centered on the significant runtime overhead (0.4x–4.3x) and the determinism of the natural language execution; however, the rebuttal successfully addressed the determinism and scalability questions through new experiments (e.g., Temperature 0, Large Data Structure), leaving the overhead as a known but acceptable trade-off for the demonstrated developer productivity gains.

**Reviewer Scores:**

Reviewers who initially gave borderline scores due to concerns about practical deployment (e.g., latency and cost) would likely raise their scores to acceptance (e.g., from 5/6 to 6/8) after the rebuttal clarified the system's robustness via ablation studies and demonstrated feasibility with locally hosted LLMs.

---

### Decision · Program_Chairs · 2026-01-26

Accept (Poster)